# Windborne migration amplifies insect-mediated pollination services

Huiru Jia[1,2], Yongqiang Liu[1], Xiaokang Li[1,3], Hui Li[1], Yunfei Pan[1], Chaoxing Hu[1,4], Xianyong Zhou[1,5], Kris AG Wyckhuys[1], Kongming Wu[1]*

[1]State Key Laboratory for Biology of Plant Diseases and Insect Pests, Institute of 8 Plant Protection, Chinese Academy of Agricultural Sciences, Beijing, China; [2]Guangdong Laboratory for Lingnan Modern Agriculture, Guangzhou, China; [3]College of Plant Protection, Shenyang Agricultural University, Shenyang, China; [4]Institute of Entomology, Guizhou Provincial Key Laboratory for Agricultural Pest, Management of the Mountainous Region, Guizhou University, Guiyang, China; [5]College of Plant Protection, Southwest University, Chongqing, China

**Abstract** Worldwide, hoverflies (Syrphidae: Diptera) provide crucial ecosystem services such as pollination and biological pest control. Although many hoverfly species exhibit migratory behavior, the spatiotemporal facets of these movement dynamics, and their ecosystem services implications are poorly understood. In this study, we use long-term (16-year) trapping records, trajectory analysis, and intrinsic (i.e., isotope, genetic, pollen) markers to describe migration patterns of the hoverfly *Episyrphus balteatus* in northern China. Our work reveals how *E. balteatus* migrate northward during spring–summer and exhibits return (long-range) migration during autumn. The extensive genetic mixing and high genetic diversity of *E. balteatus* populations underscore its adaptive capacity to environmental disturbances, for example, climate change. Pollen markers and molecular gut analysis further illuminate how *E. balteatus* visits min. 1012 flowering plant species (39 orders) over space and time. By thus delineating *E. balteatus* transregional movements and pollination networks, we advance our understanding of its migration ecology and facilitate the design of targeted strategies to conserve and enhance its ecosystem services.

*For correspondence: wukongming@caas.cn

Competing interest: The authors declare that no competing interests exist.

## Editor's evaluation

Hoverflies are a group of insects which provide crucial ecosystem services such as pollination and crop protection. Their migratory behavior in western countries is well characterized, but in eastern Asia, the annual summer monsoon provides a 'highway' of favorable winds for the airborne transport of migratory organisms, and the migration of hoverflies in this large region has not been well studied. This study addresses hoverfly migration in East Asia and its consequences using a variety of suitable methods and will be of great interest to insect migration biologists and pollination ecologists.

## Introduction

Migration plays a key role in the evolution and life history of many organisms, with insects being the most abundant, speciose, and economically important group of terrestrial migrants (*Chapman et al., 2011*). Across the globe, billions of insects annually undertake long-range movements. By transporting energy, nutrients, and other organisms between distant regions, insect migrants provide a multitude of ecosystem services and disservices (*Chapman et al., 2012*; *Hu et al., 2016*; *Satterfield et al., 2020*). Despite the important socioecological consequences of insect migration, research

has primarily centered on a handful of large-bodied charismatic species or agricultural pests (e.g., monarch butterflies, locusts; *Chapman et al., 2015*). For most other taxa, there is a critical dearth of information.

Hoverflies (Diptera: Syrphidae) are a speciose family of beneficial insects – deemed to be the second most important pollinators after bees (*Branquart and Hemptinne, 2000*; *Rader et al., 2020*). The larval stages of many hoverflies are effective predators of homopteran feeders, providing natural biological control across geographies and farming contexts (*Tenhumberg, 1995*; *Tenhumberg and Poehling, 1995*). Evidence to date suggests that hoverfly species are abundant diurnal migrants that deliver ecosystem services in both natural and man-made habitats (e.g., *David, 1951*; *Aubert and Tiefenau, 1981*; *Wotton et al., 2019*). Moreover, given that (migratory) hoverflies exhibit comparatively stable population numbers and transport pollen over long distances (*Wotton et al., 2019*), these species potentially can sustain pollination and pest control services in the face of a global insect decline (*Sánchez-Bayo and Wyckhuys, 2020*; *Powney et al., 2019*). Yet, though (long-range) dispersal is a central determinant of their survival, hoverfly migration has only been intermittently studied since the 1950s. In order to effectively conserve these organisms and to raise their contribution to (agro-)ecosystem functioning, a more in-depth understanding needs to be gained of hoverfly migration.

In recent years, several new technologies have helped to uncover insects' seasonal migration patterns and population genetic structure. Stable isotope analysis, molecular genetics, tethered flight mill assays, insect radar, and aerial trapping have all yielded insights into the migration behavior of hoverflies (*Raymond et al., 2014*; *Dällenbach et al., 2018*; *Wotton et al., 2019*; *Gao et al., 2020*). Attempts have equally been made to capture the geographical extent and ecological impacts of hoverfly migration (*Wotton et al., 2019*). Most of these studies however originate from (a small area within) Europe, while virtually no information is available from other parts of the world. Also, as the nutritional ecology of most species waits to be deciphered, little is known about hoverfly–plant associations and how those are modulated by (long-range) migration dynamics.

In China, approx. 580 hoverfly species have been described. These include the marmalade hoverfly *Episyrphus balteatus* (DeGeer) (*Li et al., 2009*), a common flower visitor in urban and agricultural settings across the Palearctic realm. Locally, (insect) migration primarily takes place within the East Asia monsoon climatic zone (*Drake and Farrow, 1988*). Owing to its geographical range, complex topography, and diverse agroecological conditions, this climatic zone constitutes an exceptional setting to study broad-scale migration dynamics of hoverfly species, for example, as compared to other parts of the globe (*Wotton et al., 2019*; *Menz et al., 2019*; *Finch and Cook, 2020*).

In northern China, numerous insect species annually undertake (two-way) migration across the Bohai Strait, influenced by the East Asian monsoon cycle (*Feng et al., 2003*). In this study, we employed a suite of novel methodologies to characterize the migration dynamics of *E. balteatus* in China. More specifically, we conducted long-term (16-year) searchlight trapping on Beihuang (BH), a small isolated island with sparse vegetation in the center of the Bohai Strait, to clarify whether *E. balteatus* engages in long-range migration and to describe the ensuing migration patterns. Second, we deployed backward trajectory analysis and stable isotope analysis to infer the *E. balteatus* migration routes and source areas. Third, we employed a population genetics approach to compare the genetic makeup and demographic history of migrant and field-collected individuals throughout China. Fourth, we described *E. balteatus* host–plant associations by identifying the pollen grains attached to hoverfly bodies. Lastly, we paired molecular gut content analysis with high-throughput sequencing (HTS) to investigate the spatiotemporal distribution of its (flower) host plants across a broad geographic range. As such, our work characterized *E. balteatus* migration behavior and captured its broader ecological relevance, for example, in terms of flower visitation networks, pollination, or natural biological control services.

## Results

### Migration dynamics

Despite the local availability of weedy host plants, intensive field surveys did not detect a presence of *E. balteatus* larvae on Beihuang (BH) island. Yet, night-time trapping consistently yielded *E. balteatus* adults from late April to October (*Figure 1A*) throughout the 16-year sampling period. Hoverfly

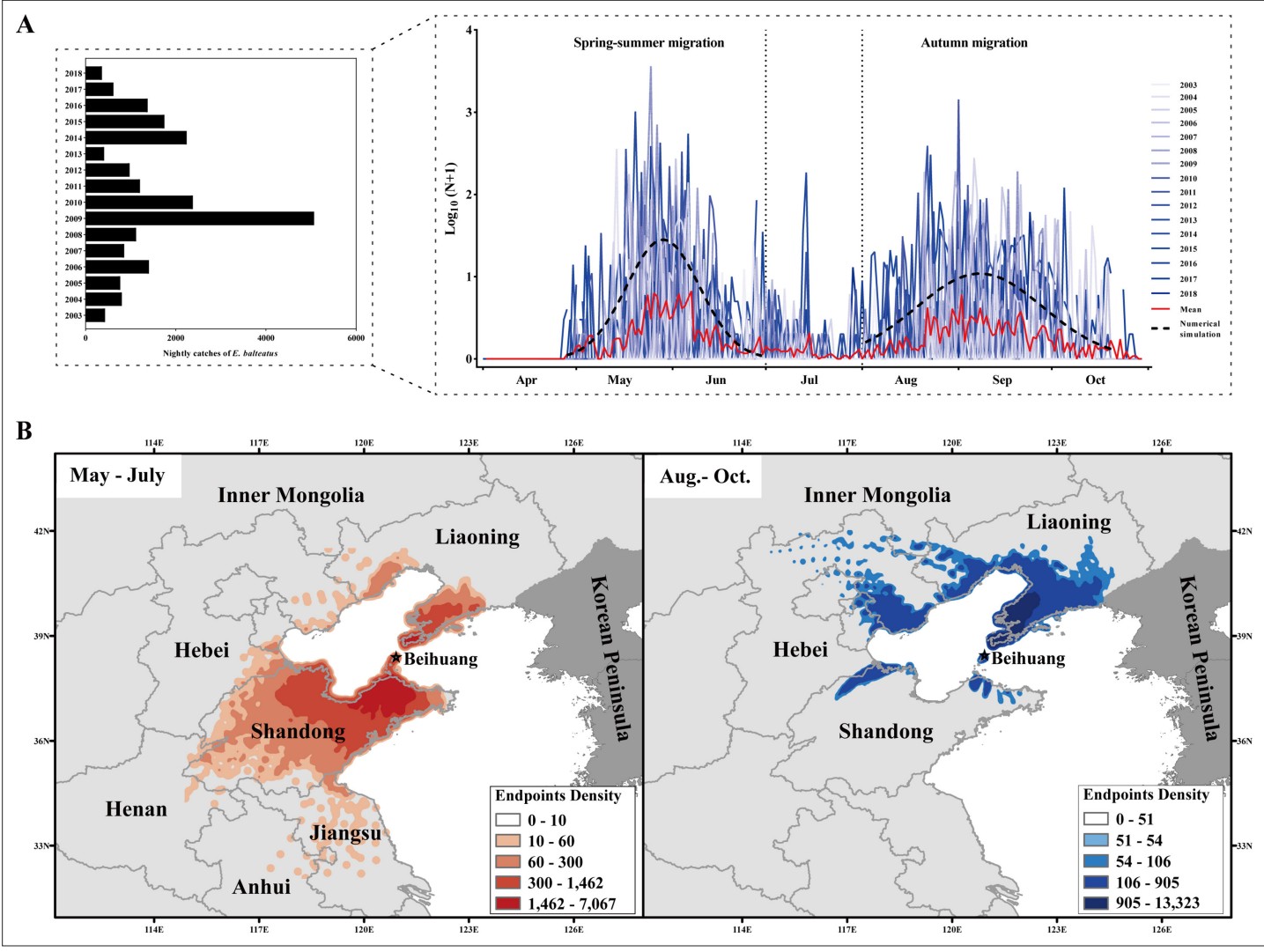

**Figure 1.** Seasonal migration patterns of *Episyrphus balteatus* in Eastern China. (**A**) Annual migration dynamics, expressed as nightly searchlight-trap catches, of *E. balteatus* on Beihuang Island (BH; Bohai Gulf, China) from April to October 2003–2018. (**B**) Endpoints of backward trajectories of BH-caught *E. balteatus* individuals during mass migration events over 2003–2018 for a 12-hr flight duration. Darker colors indicate a higher density of endpoints at a particular location. The left panel (i.e., orange colors) reveals the possible source areas of late spring and summer immigrants, while the right panel (i.e., blue colors) indicates those of autumn immigrants.

The online version of this article includes the following source data and figure supplement(s) for figure 1:

**Source data 1.** Nightly catches of E. balteatus in the searchlight trap on BH from April to October 2003–2018.

**Figure supplement 1.** Analysis of variance on the number of *E. balteatus* captured in the searchlight trap on BH Island from May to October 2003–2018 by comparing EMMs.

abundance (or trap capture rate) showed important annual variation (Marginal $R^2$ = 0.163, $\chi^2$ = 75.4, df = 15, p < 0.001) (*Figure 1—figure supplement 1*), with peak population sizes of 5068 and 2381 individuals in 2009 and 2010, respectively. Overall, *E. balteatus* trap capture rate exhibited a bimodal pattern with peak abundance from May to June and August to September, thus comprising two distinct migration stages. On an annual basis, *E. balteatus* migration covered a period of 151.2 ± 17.9 days (*Supplementary file 1e*).

Next, the possible origin of *E. balteatus* migrants on BH was identified through backward trajectory analysis using the HYSPLIT model. For 'mass migration' events over 2003–2018 (*Supplementary file 1a*), spring–summer migrants primarily originated in southern areas, while autumn migrants arrived from northern areas. During spring–summer, a total of 9069 valid endpoints endpoints were identified and more than 90% endpoints where located south of BH, that is, in Shandong, Jiangsu, Henan, and

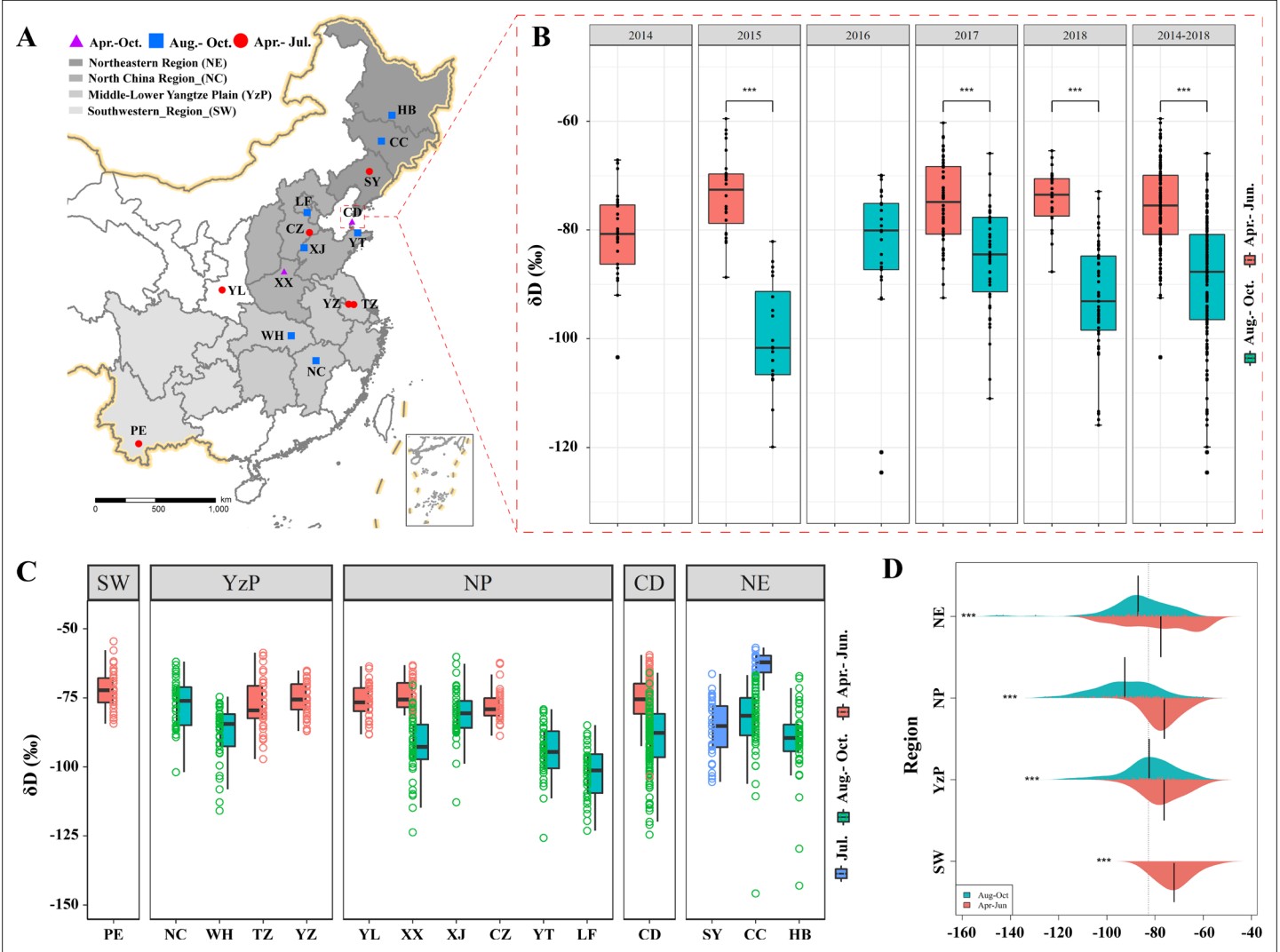

**Figure 2.** Stable hydrogen isotope analysis of *E. balteatus*. (**A**) Sampling locations. (**B**) Seasonal incidence of mean δD values in the wings of migratory *E. balteatus* adults, as recorded for different migration stages over 2014–2018 on Beihuang Island (BH). Double asterisks (***) indicate a statistically significant difference (p < 0.01, Wilcoxon test). (**C, D**) Seasonal incidence of mean δD values in the wings of wild-caught individuals from each sampling location or geographic region during April to October 2017–2018, respectively. Locations are ranked as per their longitudinal position. PE, Puer; NC, Nanchang; WH, Wuhan; TZ, Taizhou; YZ, Yangzhou; YL,Yangling; XX, Xinxiang; XJ, Xiajin; CZ,Cangzhou; YT, Yantai; LF, Langfang; CD, Changdao; SY, Shenyang; CC, Changchun; HB, Harbin.

Anhui provinces; during autumn, of 4945 valid endpoints, 85% endpoints were distributed in areas north of BH such as Liaoning (79.78%) likely acted as key source areas (*Figure 1B*; *Supplementary file 1f*).

To corroborate the above patterns, wings of BH migrants and wild-caught individuals from sites across China were subject to hydrogen isotope analysis. On BH, greater variability in δD values was recorded for autumn versus spring–summer migrants over the entire study period (Wilcoxon rank-sum test; $W = 11.93$, df = 254.13, p < 0.0001) (*Figure 2B*). By comparing the above δD values with its established precipitation gradient, source areas of *E. balteatus* migrants were identified. Considering how δD values in field-collected populations divert from latitudinal gradients (*Figure 2C*), seasonal differences became apparent upon a geographical grouping of *E. balteatus* populations. Overall, spring-caught adults showed higher δD values than those captured during autumn (Wilcoxon rank-sum test; NE subgroup: $W = 3232.5$, p < 0.001; NP: $W = 13804$, p < 0.001; YzP: $W = 3440$, p < 0.001) (*Figure 2D*). Isotope patterns reflected how hoverfly adults engage in bidirectional migration over hundreds of kilometers.

**Table 1.** Genetic diversity indices of 18 *E. balteatus* populations based on Cytb and 18S-28S rRNA gene.

For each population (and sampling location), the following metrics are reported: *N*, sample size; *S*, number of segregating sites; *H*, number of haplotypes; Hd, haplotype diversity; *K*, average number of differences; Pi, nucleotide diversity. For the site names see *Figure 2*.

| Site code | CytB | | | | | | | 18S-28S rRNA | | | | | | |
| | N | S | H | Hd | K | Pi | PiJC | N | S | H | Hd | K | Pi | PiJC |
| --- | --- | --- | --- | --- | --- | --- | --- | --- | --- | --- | --- | --- | --- | --- |
| LF | 23 | 37 | 7 | 0.46245 | 3.92095 | 0.00536 | 0.00543 | 12 | 25 | 8 | 0.84848 | 5.51515 | 0.00452 | 0.00454 |
| XX | 42 | 6 | 5 | 0.33682 | 0.45296 | 0.00062 | 0.00062 | 16 | 31 | 10 | 0.825 | 4.79167 | 0.00392 | 0.00394 |
| HB | 23 | 1 | 2 | 0.08696 | 0.08696 | 0.00012 | 0.00012 | 18 | 28 | 10 | 0.81046 | 5.60784 | 0.00459 | 0.00461 |
| WH | 34 | 10 | 5 | 0.2246 | 0.58824 | 0.0008 | 0.00081 | 12 | 32 | 9 | 0.90909 | 6.90909 | 0.00566 | 0.00569 |
| CC | 16 | 28 | 7 | 0.625 | 3.5 | 0.00478 | 0.00482 | 11 | 22 | 10 | 0.98182 | 5.63636 | 0.00462 | 0.00464 |
| NC | 21 | 12 | 5 | 0.35238 | 1.65714 | 0.00226 | 0.00228 | 15 | 45 | 12 | 0.94286 | 9.61905 | 0.00788 | 0.00793 |
| SY | 45 | 16 | 6 | 0.25051 | 0.75354 | 0.00103 | 0.00104 | 21 | 49 | 18 | 0.98095 | 9.2619 | 0.00759 | 0.00764 |
| CF | 27 | 21 | 8 | 0.45869 | 2.02849 | 0.00277 | 0.00279 | 16 | 39 | 16 | 1 | 9.00833 | 0.00738 | 0.00742 |
| XJ | 25 | 1 | 2 | 0.08 | 0.08 | 0.00011 | 0.00011 | 18 | 31 | 13 | 0.95425 | 6.71895 | 0.0055 | 0.00553 |
| YT | 29 | 22 | 5 | 0.31773 | 1.95074 | 0.00266 | 0.00268 | 16 | 34 | 12 | 0.95833 | 7.8 | 0.00639 | 0.00643 |
| YP | 18 | 8 | 3 | 0.21569 | 0.88889 | 0.00121 | 0.00122 | 8 | 38 | 8 | 1 | 12.89286 | 0.01056 | 0.01064 |
| YL | 30 | 4 | 6 | 0.31034 | 0.33103 | 0.00045 | 0.00045 | 14 | 35 | 14 | 1 | 8.56044 | 0.00701 | 0.00705 |
| TZ | 28 | 2 | 3 | 0.14021 | 0.14286 | 0.0002 | 0.0002 | N/A | N/A | N/A | N/A | N/A | N/A | N/A |
| PE | 18 | 7 | 4 | 0.47059 | 1.79739 | 0.00246 | 0.00247 | 11 | 98 | 11 | 1 | 22.96364 | 0.01881 | 0.01922 |
| YZ | 31 | 4 | 5 | 0.24516 | 0.25806 | 0.00035 | 0.00035 | 10 | 72 | 9 | 0.97778 | 15.62222 | 0.01279 | 0.01309 |
| SH | 7 | 8 | 5 | 0.85714 | 2.28571 | 0.00312 | 0.00313 | N/A | N/A | N/A | N/A | N/A | N/A | N/A |
| CDI | 52 | 43 | 16 | 0.52413 | 2.30166 | 0.00314 | 0.00316 | 25 | 103 | 21 | 0.97667 | 16.9 | 0.01384 | 0.01413 |
| CDII | 61 | 48 | 17 | 0.48251 | 1.70055 | 0.00232 | 0.00234 | 37 | 65 | 28 | 0.98198 | 9.15165 | 0.0075 | 0.00754 |

## Population genetics

To gain genetic evidence of its regional migration, we described *E. balteatus* genetic diversity and population structure using one mitochondrial DNA gene (Cytb) and two nuclear DNA genes (i.e., 18s rRNA, 28s rRNA). Upon analysis of 670 field-collected specimens and 133 light-trapped individuals (representing a respective 16 and 2 populations), high haplotype diversity and low nucleotide diversity was recorded (*Table 1*). Based upon Cytb sequences, 83 haplotypes were identified among 530 individuals, with haplotype diversity (Hd) ranging from 0.0800 (XJ) to 0.857 (SH) (total = 0.351) and nucleotide diversity ($\pi$) from 0.000110 to 0.00536 (total = 0.00178), respectively. Conversely, the concatenated nuclear gene possesses improbably high haplotype diversity and low nucleotide diversity. Up to 145 haplotypes were detected among 260 individuals with Hd ranging from 0.810 to 1 (total = 0.961) and $\pi$ ranging from 0.00392 to 0.01384 (total = 0.000008).

Though phylogenetic analyses showed four and five distinct clades among the Cytb and nuclear haplotypes, haplotype and geographical origins were not linked (*Figure 3—figure supplement 1*). Median-joining also did not reveal geographical clustering. Instead, a star-like pattern was displayed with the most common, ancient haplotypes in the center (*Figure 3—figure supplement 2*). Most haplotypes were unique to individuals and populations, while only 9 (out of 83 Cytb haplotypes) and 30 (out of 145 nuclear haplotypes) were shared. In each population, shared haplotypes occurred at 63–100% frequencies for *Cytb* and 33–88% for nuclear genes (*Figure 3A, B*). Only 1.8% and 11.1% of genetic variation for the respective *Cytb* and nuclear gene could be attributed to variation among populations (*Supplementary file 1g*). Low pairwise $F_{ST}$ values between different localities equally reflected low levels of genetic differentiation (*Figure 3—figure supplement 3*). When describing effective population sizes and migration rates between (geographically grouped) populations, high levels of interpopulation gene flow were recorded (*Figure 3C, D*). Migration between *E. balteatus* populations was asymmetrical and exhibited a general migration trend toward the Yangtze basin,

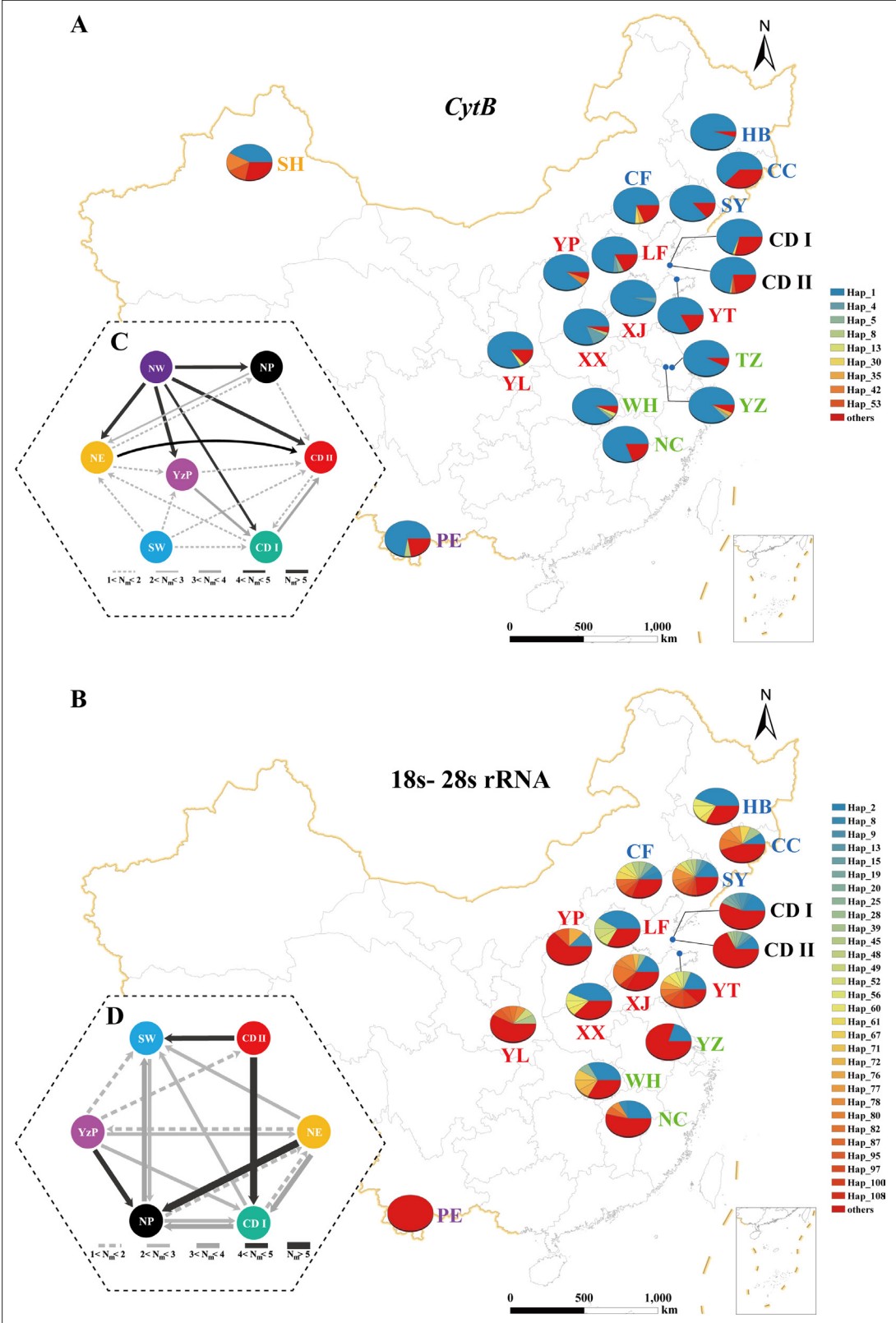

**Figure 3.** Population genetic structure of *E. balteatus* populations sampled in 18 locations across China. Analyses are either based on one mitochondrial *Cytb* gene (upper panels, i.e., **A, C**) or on combined 18s and 28s rRNA nuclear genes (lower panels, i.e., **B, D**). (A, B), Spatial distribution of *E. balteatus* haplotypes. At each given location, a pie chart shows the proportional abundance of haplotypes. (C, D) Migration dynamics of different

*Figure 3 continued on next page*

*Figure 3 continued*

populations as estimated through MIGRATE-N. Individual arrows represent the prevailing migration direction and arrow thickness is proportional to the number of migrants. SW, Southwestern Region; YzP, Yangtze Plain; NP, Northern Region; and NE, Northeastern Region, as defined in *Figure 2*.

The online version of this article includes the following figure supplement(s) for figure 3:

**Figure supplement 1.** Neighbor-joining (NJ) phylogenetic trees of haplotypes, in which different clades are depicted with different colors.

**Figure supplement 2.** Median-joining haplotype network of the CytB (**a**) and 18S-28S rRNA gene (**b**).

**Figure supplement 3.** Heatmap diagram of genetic differentiation coefficient (*Fst*) between populations based on the *CytB* (**a**) and combined 18S-28S rRNA (b) gene.

**Figure supplement 4.** Mismatch distribution of the *Cytb* and18S-28S rRNA gene in *E. balteatus* populations.

with north > center, south > center, and west > east as dominant directions. Migration thus enables genetic mixing among *E. balteatus* populations from remote origins. Given the negative Tajima's $D$ and Fu's $F_S$ (*Cytb*: Tajima's $D = -2.72034$, Fu's $F_S = -27.4859$; concatenated 18S-28S rRNA: Tajima's $D = -2.33710$, Fu's $F_S = -34.456$) and unimodal distribution for both markers (***Figure 3—figure supplement 4***), *E. balteatus* populations possibly experienced recent population expansion.

## Palynological analysis

Morphological characteristics and barcode markers were used to identify pollen species attached to migrating *E. balteatus* adults, and to infer the associated movement and flower visitation patterns. Among 1014 BH adult migrants collected during 2015–2018, 32% had pollen grains adhering to the body surface. Using a combination of DNA sequences, pollen morphology and plant distribution records, a total of 46 pollen species representing at least 42 plant genera and 26 families were identified (***Figure 4***; ***Supplementary file 1h***). Out of these, 10 were identified to genus level and the remained to species level. Few adults carried pollen from multiple plant species. Pollen-bearing plants mainly pertained to Asteraceae (12), Moraceae (4), and Celastraceae (3), and were primarily herbaceous as compared to woody plants ($\chi^2 = 112.26$, df = 1, p < 0.001). It should be noted that most plant hosts were identified from one single hoverfly, with only few adults carrying pollen from different plant species.

Furthermore, temporal patterns were recorded in the pollen adherence ratio, with the respective highest and lowest levels recorded in October (50%) and April (17%). Also, the identity of pollen grains that adhered to *E. balteatus* bodies equally exhibited seasonal variation. Specifically, 29 pollen species were identified from spring–summer migrants, with *Taraxacum mongolicum*, *Ailanthus altissima*, *Amorpha fruticose*, and *Chenopodium giganteum* the most common plant hosts (i.e., accounting for a 72% total carrier rate). Conversely, 17 pollen species were identified from autumn migrants, with *Artemisia* L., *Chrysanthemum zawadskii* and *Ambrosia trifida* accounting for a 82.2% total carrier rate (***Supplementary file 1c***).

Lastly, specific plant taxa were associated with certain migration stages. Plants endemic to certain ecological zones helped to pinpoint migration origins (***Jones and Jones, 2001***). During spring–summer, the presence of *Citrus* L., *Sedum japonicum*, or *Euonymus myrianthus* endemic to central and southern China hinted at migration origins in south-central China. Conversely, during autumn, the presence of *C. zawadskii* mirrored potential migration origins in northeastern China (***Figure 5***). These results further reflected northward *E. balteatus* migration flows during spring with return movements in autumn.

## Molecular gut content analysis

Under laboratory conditions, a polymerase chain reaction PCR-based assay was developed to identify the plant species that are consumed by *E. balteatus*. Using DNA extracts from excised syrphid abdomens and ITS2-targeted plant primers, we successfully amplified the expected band with specific size of all extracts. BLAST analysis confirmed that sequencing products were indeed the three target plant species (*M. sativa*, *H. scandens*, and *H. annuus*) that were consumed by *E. balteatus* adults.

Additionally, DNA of all three ingested plants could be detected (at high frequency) in *E. balteatus* guts for up to 9-day postfeeding, the maximum digestion time in our experiments. Moreover, qPCR methods further yielded standard curves to quantify the degradation of the plant DNA through digestion, and a negative exponential equation to compute the time to reach a 50% detection probability

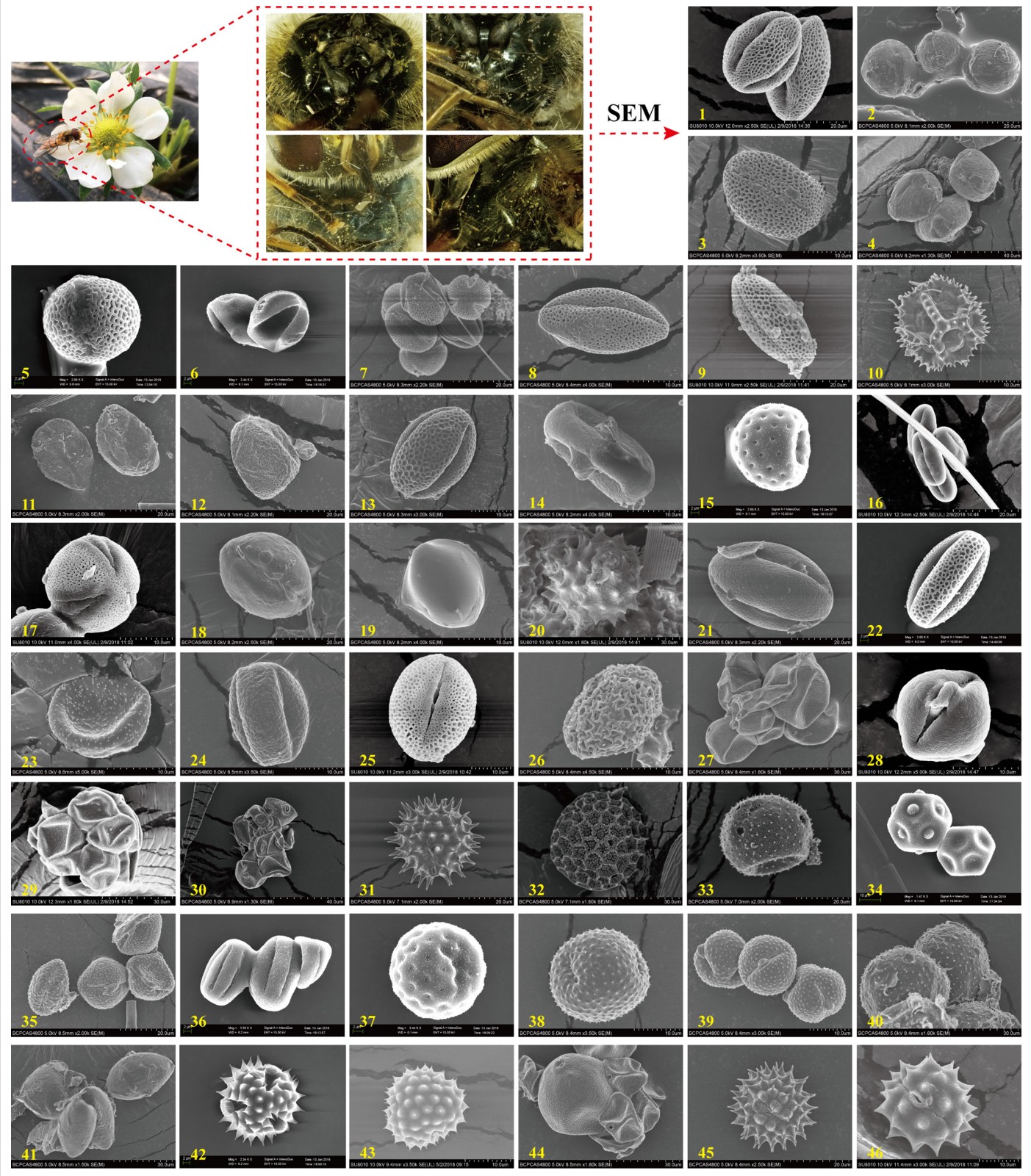

**Figure 4.** Scanning electron microscopy (SEM) microphotographs of pollen grains attached to *E. balteatus* migrants on BH during 2014–2018. 1. *Ailanthus altissima*; 2. *Cotinus coggygria*; 3. *Forsythia suspensa*; 4. *Prunus avium*; 5. *Brassica* L.; 6. *Morus alba*; 7. *Citrus sinensis*; 8. *Descurainia sophia*; 9. *Euonymus* L.; 10. *Taraxacum* L.; 11. *Sedum japonicum*; 12. *Populus cathayana*; 13. *Celastrus orbiculatus*; 14. *Daucus carota*; 15. *Chenopodium* L.; 16. *Castanea mollissima*; 17. *Amorpha fruticosa*; 18. *Diospyros lotus*; 19. *Ziziphus jujuba*; 20. *Cirsium setosum*; 21. *Neoshirakia japonica*; 22. *Flueggea* L.; 23.

*Figure 4 continued on next page*

*Figure 4 continued*

*Maclura pomifera*; 24. *Rumex* L.; 25. *Euonymus* L.; 26. *Schisandra chinensis*; 27. *Eleusine indica*; 28. *Actinidia kolomikta*; 29. *Cannabis sativa*; 30. *Humulus scandens*; 31. *Helianthus annuus*; 32. *Persicaria orientalis*; 33. *Adenophora trachelioides*; 34. *Gypsophila paniculata*; 35. *Artemisia* L.; 36. *Rubia cordifolia*; 37. *Rubia cordifolia*; 38. *Artemisia* L.; 39. *Artemisia* L.; 40. *Artemisia* L.; 41. *Allium tuberosum*; 42. *Tripolium vulgare*; 43. *Ambrosia trifida*; 44. *Sorghum bicolor*; 45. *Aster tataricus*; 46. *Chrysanthemum zawadskii*. The scale bar is shown on the bottom of each photograph.

(*Figure 6*). Detection times ranged between 5.14 and 5.79 hr for the three plant species. Once more than 8.9 copy numbers/µl of ingested plant DNA were present in *E. balteatus* guts, amplification product could also be visualized by ethidium bromide staining.

## Spatiotemporal dietary shifts

By pairing gut content analysis with HTS (Illumina MiSeq), (floral) diet profiles of 180 light-trapped *E. balteatus* migrants on BH and 436 field-collected individuals at 19 different locations in China were elucidated. MiSeq paired-end sequencing of the *ITS2* gene yielded 3,348,773 raw reads from 616 samples. After quality filtering and reference-based chimera removal, a total of 2,952,448 sequences remained, ranging from 100 to 28,582 sequences per sample. These valid reads were clustered into Operational Taxonomic Units (OTUs) for further analysis (*Supplementary file 2*). According to the taxonomic assignment, almost all sequences were identified at least to the genus level. Overall, 1012 plant species belonging to 39 orders, 91 families, and 429 genera were identified, with *Asteraceae* (84), *Poaceae* (37), *Apiaceae* (21), and *Fabaceae* (21) (count at genus level) the dominant families

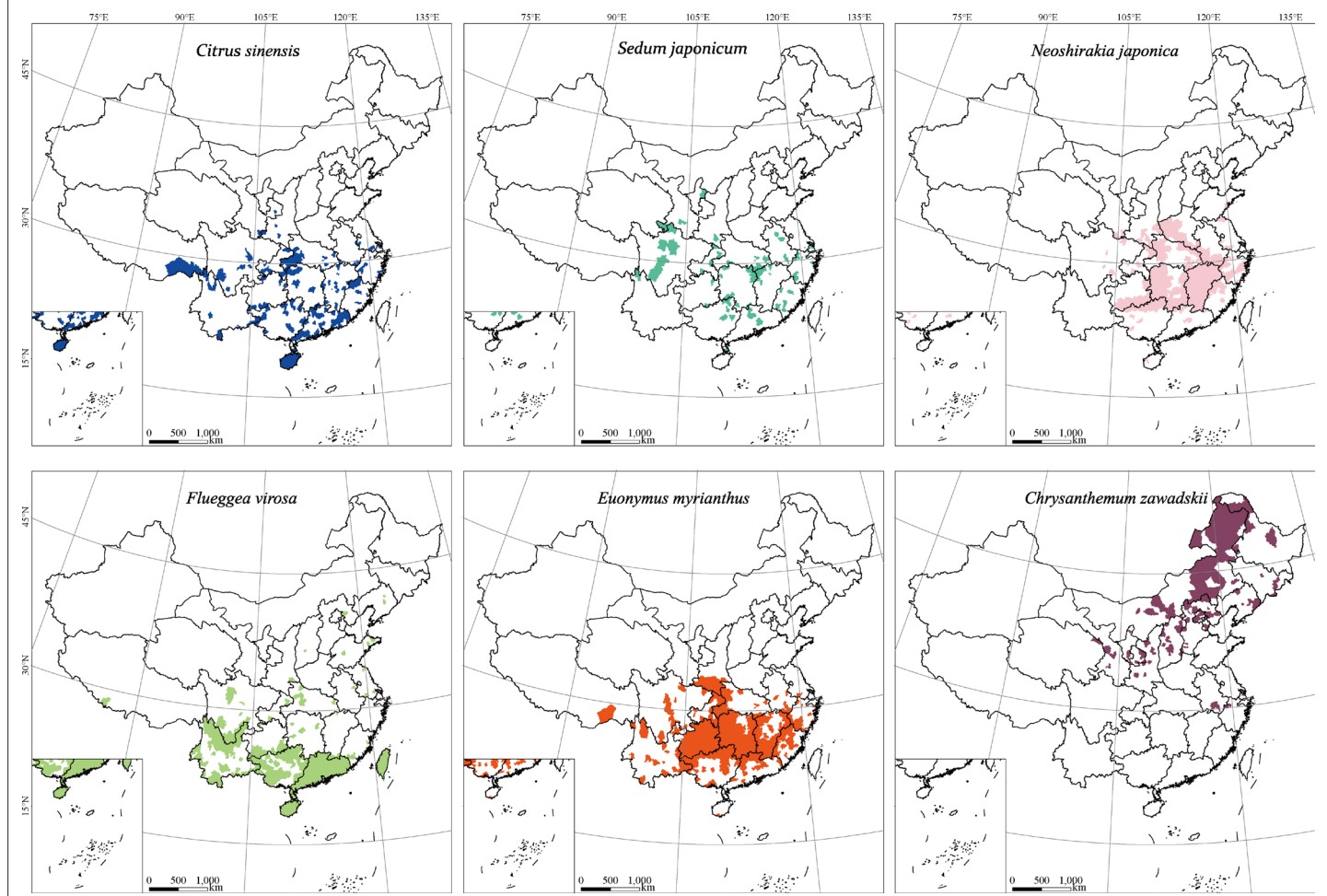

**Figure 5.** Geographical distribution of pollen-bearing plant species that help delineate *E. balteatus* migration patterns. Maps show the (district-level) geographical distribution of individual plant species.

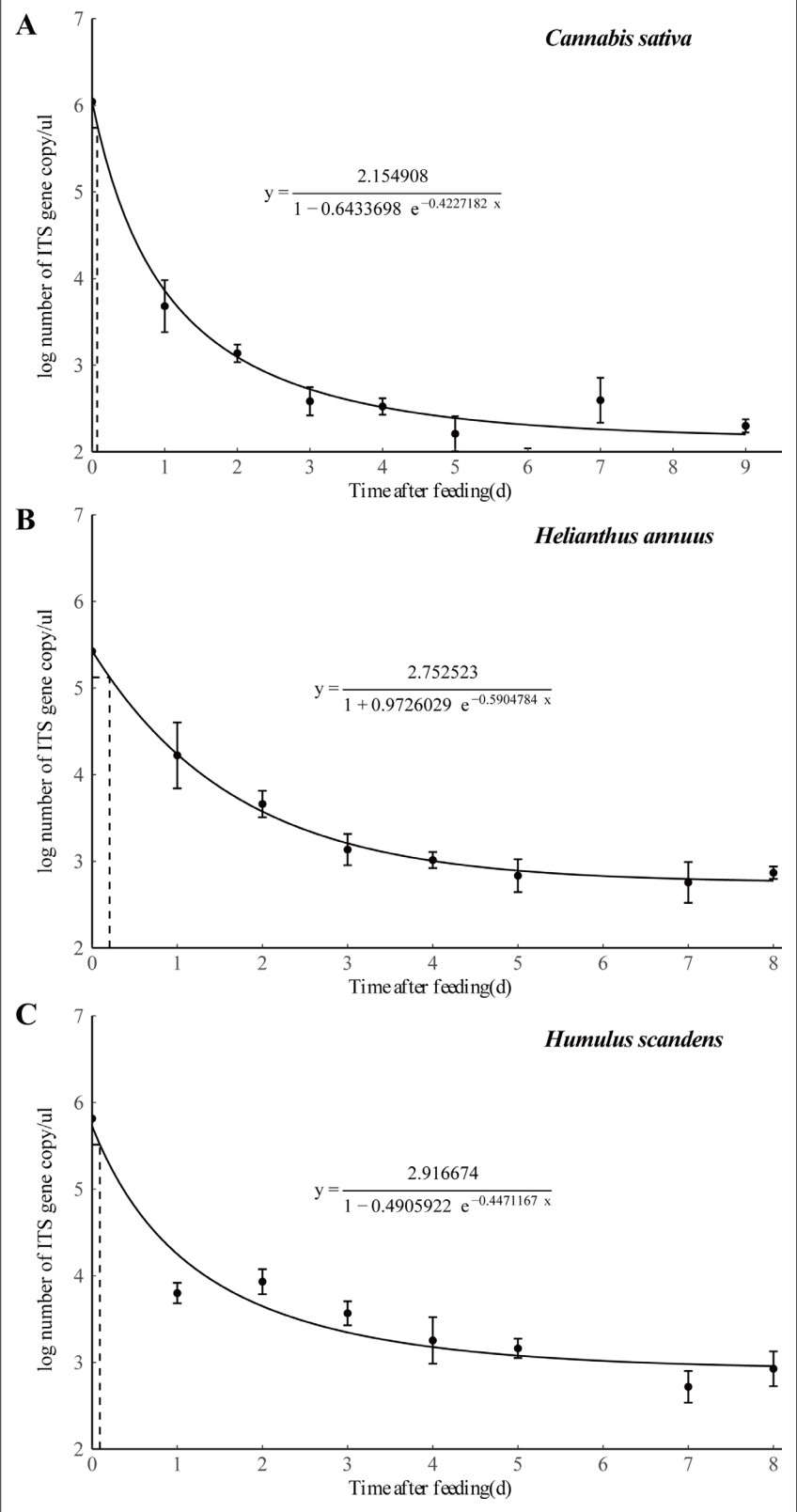

**Figure 6.** Detection of three different plant DNA in the guts of *E. balteatus* adults at different times after ingestion by qPCR analysis.

Error bars at each point on the curves represent the standard error of replicates.

*Figure 6 continued on next page*

*Figure 6 continued*

The online version of this article includes the following source data and figure supplement(s) for figure 6:

**Figure supplement 1.** The respective standard curve equations of the three plants in the feeding experiments.

**Figure supplement 1—source data 1.** qPCR analysis of three different plant DNA in the guts of E.balteatus adults at different times after ingestion.

accounting for 38.0% of all plant taxa (**Supplementary file 3**). Further analysis of plant characteristics of identified plant taxa revealed important similarities with the above palynological analysis, in which more Angiosperm and Dicotyledon plants were recorded than Gymnosperms ($\chi^2$ = 393.76, df = 1, p < 0.0001) or Monocotyledons ($\chi^2$ = 253.04, df = 1, p < 0.0001). Woody plant hosts were more common than herbaceous ones ($\chi^2$ = 61.94, df = 1, p < 0.0001) (**Figure 7**). These results indicated the hoverflies might have visited herbaceous plants more often than woody plants.

The diversity and abundance of host plant communities varied across geographical sites and sampling times (**Figure 7—figure supplements 1 and 2**). In particular, significantly more plant taxa were recorded in the BH migratory population (36 orders, 76 families, and 320 genera) as compared to field-collected populations. Moreover, Venn diagram displayed that only a small number of plant taxa were unique to particular (geographical) populations at both the phylum and genus levels, most of the these were widespread among two or more regions, and there are several taxa that included six families *Apiaceae*, *Poaceae*, *Asteraceae*, *Brassicaceae*, *Fabaceae*, *Ranunculaceae*, and one genera *Brassica* observed in all of the populations across different geographical sites and sampling times (**Figure 7—figure supplement 3a, b**). The results of hierarchical cluster analysis revealed that the overall diet composition of this species exhibited weak temporal differences upon a spatiotemporal grouping of *E. balteatus* populations at the genus level (**Figure 7—figure supplement 4**). These results strongly indicated that geographical location thus only affected pollen transport networks to small extent. Importantly, the host plant composition of the BH migratory population almost equals the sum of 19 field-collected populations (**Figure 7—figure supplement 3c, d**), which suggested the migratory behavior of some insects has important role in shaping flower visitation network structures.

Next, upon analyzing host plants derived from light-trapped *E. balteatus* migrants on BH (36 orders, 76 families, and 320 genera), we further assessed how sampling time did affect the primary (plant) foraging resource. The results showed that the overall plant community composition exhibited important within-year variability, as reflected by the host-associated community separation visualized in the principal coordinates analysis (PCA) profile and the significance test based on analysis of similarities (ANOSIM; p < 0.001; **Figure 7—figure supplement 5a**). Moreover, Venn diagrams indeed displayed a certain number of unique plant species in all *E. balteatus* groups, with a clear shift in plant taxa over time (**Figure 7—figure supplement 5b**). Based on the average relative abundance, *Asteraceae* was the most important plant families during the whole study period (**Figure 7—figure supplement 6a**). At the genus level, during spring (April to May), the genera *Pinus* and *Capsella* are the important food resources; during summer (June to August), *Flueggea* and Maclura became more common pollen hosts; during autumn (September–October), *Ajania* and *Artemisia* was the most represented species (**Figure 7—figure supplement 6b**), the importance of these main plant taxa during specific seasons was also confirmed by random forest machine learning (**Figure 7—figure supplement 7**). Also, in addition to the above five plants that are informative of migration identified by pollen analysis, more geographically confined species such as *Acer pictum*, *Pittosporum truncatum*, *Pseudotsuga sinensis*, and *Xylosma congesta*, have been identified by applying this method (**Figure 7—figure supplement 8**).

## Discussion

Several European studies have unveiled how hoverflies engage in long-distance migration, yet little is known about such phenomenon in other parts of the world. In this study, we relied upon different analytical approaches to delineate (long-range) dispersal patterns of the widely distributed *E. balteatus* in Asia. Specifically, combining spatially explicit data from searchlight trapping, trajectory analysis, molecular palynology, and metabarcoding, we showed how *E. balteatus* annually engages in bidirectional migration over hundreds of kilometers. Our study further revealed high genetic diversity and extensive genetic mixing at a continental scale, hinting at the superior adaptability and plasticity

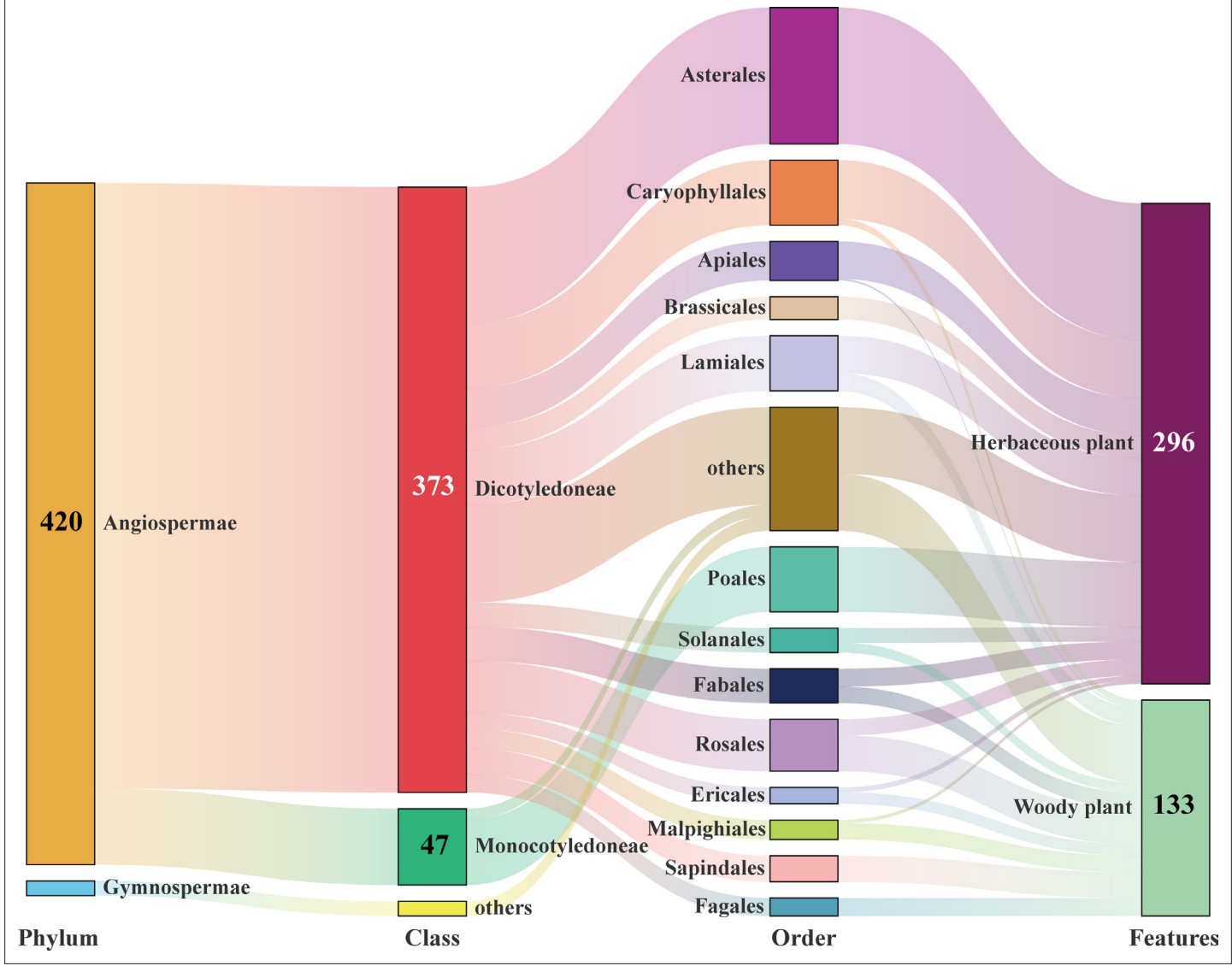

**Figure 7.** Sankey diagram summarizing the host plants of *E. balteatus*, as identified via DNA metabarcoding (i.e., for BH migrant individuals) or DNA-based gut content analysis (i.e., for field-collected individuals from 19 sites).
Taxonomic structure is shown for 429 plant genera, with individual numbers referring to the number of counts of particular taxa.

The online version of this article includes the following source data and figure supplement(s) for figure 7:

**Source data 1.** List of the host plants of E. balteatus, as identified via DNA metabarcoding (i.e., for BH migrant individuals) or DNA-based gut content analysis (i.e., for field-collected individuals from 19 sites).

**Figure supplement 1.** Host plants of *E. balteatus*, as identified through DNA metabarcoding (i.e., for BH migrant individuals) or DNA-based gut content analysis (i.e., for field-collected individuals from 19 sites).

**Figure supplement 2.** Relative abundance profiles of the 10 prevailing host plants of adult *E.*

**Figure supplement 2—source data 1.** Host plants across different regions, with plant hosts specified at the family or genus level.

**Figure supplement 3.** Spatial variability in *E. balteatus* host plant usage at 20 sites across China, as inferred through molecular gut content analysis.

**Figure supplement 4.** Hierarchical clustering tree for plant hosts among different populations, as specified at the genus level.

**Figure supplement 5.** Temporal shifts in host plant usage of *E. balteatus* migrants on BH during different migrating seasons, as determined by molecular gut content analysis.

*Figure 7 continued on next page*

*Figure 7 continued*

**Figure supplement 6.** Temporal shifts in host plant usage for *E. balteatus* migrants on BH, as determined through molecular gut content analysis.

**Figure supplement 7.** Top 30 plant genera prioritized by random forest analysis and ranked by the mean decrease in accuracy.

**Figure supplement 8.** Geographical distribution of plant species that help delineate *E. balteatus* migration patterns.

---

of this species. Pollen marker and next-generation sequencing NGS-based gut content approaches further illuminated how *E. balteatus* exploits a spatiotemporally diverse community of pollen-bearing plants. In view of the unrelenting global environmental degradation and the precipitous decline of insect populations, our work helps to guide interventions to conserve this beneficial insect species and to safeguard its vital ecosystem services.

BH island has proven to be a highly suitable site to monitor insect migration in eastern Asia, because of its unique geographical location and local habitat condition. BH island is positioned within a major migration corridor at considerable distance from mainland China, and contains no arable land or extensive vegetation. From 2003 onwards, BH has lent itself to decipher the migration behavior of more than 100 insect species (e.g., *Guo et al., 2020*), including agricultural pests such as *Helicoverpa armigera* and natural enemies such as the dragonfly *Pantala flavescens* (*Feng et al., 2009*; *Cao et al., 2018*). In the present study, drawing upon (BH-specific) light-trapping records, we demonstrate how *E. balteatus* regularly migrates across the Bohai Gulf from late April to October each year. To our best knowledge, this constitutes the first evidence for transregional migration of hoverflies in eastern Asia. Meanwhile, its population dynamics are in accordance to those of other hoverfly species in Europe and elsewhere (e.g., *Gao et al., 2020*).

As equally noted for other migratory insects (e.g., *Guo et al., 2020*; *Stefanescu et al., 2013*), important inter- and intraannual variation occurred in *E. balteatus* trap catches. Such variation can possibly be ascribed to agroclimatic conditions or biotic factors, for example, fluctuating prey abundance (*Drake and Farrow, 1988*; *Hu et al., 2021*). In our study, periodic aphid population outbreaks in mainland China likely shape *E. balteatus* migration patterns. For aphidophagous hoverflies such as *E. balteatus*, adult females deposit their eggs near aphid colonies where larvae subsequently feed until pupation; hoverfly abundance is thus inherently shaped by aphid densities (*Honěk, 2009*). Also, pest and natural enemy population are regularly coupled (e.g., *Feng et al., 2003*; *Guo et al., 2020*) and even exhibit multiyear oscillation cycles, for example, as evidenced for the soybean aphid in North America (*Rhainds et al., 2010*). Earlier work has demonstrated such pattern for the hoverfly *Eupeodes corollae* and its aphid prey (*Svensson and Janzon, 1984*), in which aphid infestation pressure potentially acts as a migration trigger. As such, (broad-scale) aphid population outbreaks may induce mass hoverfly migration events, for example, as recorded during 2009. If confirmed, hoverfly abundance levels can thus provide an 'early warning' of pest population build-up and help target interventions in arable crops. Yet, further in-depth study is required to investigate its relative value for (national, multicountry) pest surveillance programs.

Our study illuminates several key facets of hoverfly migration. Hoverfly species are believed to be diurnal fliers that exploit (high-altitude) wind currents for their migratory movements (*David, 1951*; *Menz et al., 2019*). In our study however, searchlight trapping showed how *E. balteatus* engages in night-time flight when migrating over extensive waterbodies. Our work challenges the prevailing thought, and may hint at how the day-active *E. balteatus* employs night-time migration in a similar way as certain dragonfly or beetle species (*Feng et al., 2006*; *Anderson, 2009*). On the other hand, *E. balteatus* individuals that initiated transmaritime flight during daytime may also be unable to land at sunset and thus continue their (nighttime) journey over sea. Further field observations or manipulative assays (e.g., laboratory flight mill experiments; *Zhang et al., 2009*) could clarify the existence of a 'dual' migration strategy. These studies are especially important given its ecological implications, for example, in terms of pollination or pest control.

An unambiguous delineation of migration pathways and source areas is crucial to define (spatially explicit) conservation or pest management interventions (*Chapman et al., 2015*). However, insects' flight behavior, life span and minute body size impede the use of conventional methods, for example, capture-mark-recapture or remote sensing (*Raymond et al., 2014*). Alternatively, recent and increasing advances in new tracking techniques and approaches such as endogenous markers (i.e.,

stable isotopes) can overcome these difficulties and brought useful insights into this issue (*Chapman et al., 2010*; *Hobson and Wassenaar, 2019*). In our study, by coupling HYSPLIT backward trajectory analysis with stable isotope measurements, we were able to reliably identify the migration origin and bidirectional migration patterns of *E. balteatus* over hundreds of kilometers. Specifically, we showed how *E. balteatus* migrates between agriculture-dominated regions of Northern China and Northeast China. Hoverflies thus disperse northwards in prevailing southerly winds during spring–summer (i.e., April to June), followed by a return migration in prevailing northerly winds during autumn (August to October). Similar achievements have been made for other insects in the study area (*Cao et al., 2018*; *Hu et al., 2017*) but also for migratory hoverflies in other parts of the globe (*Raymond et al., 2013*; *Gao et al., 2020*). Yet, we were unable to demarcate the exact migration routes of captured specimens or the (ephemeral, patchy) habitats that are exploited by *E. balteatus* spring migrants. This might be possible once baseline hydrogen isotope data become available for local populations or when other isotopes (C, N, and Sr) are added. Hence, follow-up studies are needed to generate finer-resolution migration patterns. Nonetheless, considering how migrant hoverflies involve in the transport of energy, nutrients and biomass, pest regulation, and pollen transfer at a macroscale (*Wotton et al., 2019*), our work has major implications for environmental preservation and sustainable agrifood production in eastern Asia. This is particularly relevant as the study region comprises some of China's main farming areas where a wide range of (aphid-affected, hoverfly-pollinated) crops are cultivated.

Population genetics studies are routinely used to assess adaptive capacity of different organisms including insects, but can also characterize migratory movements through an ecoevolutionary lens (*Kim and Sappington, 2013*). Here, although there are some differences between the *Cytb* and concatenated nuclear gene, all our population genomics analyses of field-caught *E. balteatus* from different geographical locations and BH migrants uncovered high haplotype diversity and low nucleotide diversity; a distinguishing feature of migratory species. Meanwhile, the lack of a pronounced genetic differentiation and genetic structure has been recorded for other hoverfly populations (*Hondelmann et al., 2005*; *Raymond et al., 2013*) and reflects how migration enables (interpopulation) genetic mixing over broad geographical scales. These assays also help to delineate *E. balteatus* migration routes and origins. Specifically, haplotype sharing, phylogenetic tree and gene flow analysis all revealed that the BH migratory population, field-caught populations from Northern China and Northeast China have close relationship. These findings lend further support to trajectory analyses. Additionally, the high genetic diversity and low genetic structuring reveal how *E. balteatus* likely possesses superior colonization abilities and a high adaptive potential under environmental change. As noted by *Hondelmann et al., 2005* and *Raymond et al., 2013*, *E. balteatus* exhibits high levels of genetic diversity compared to other mobile insects, common crop pests or invasive species (*Facon et al., 2011*; *Hsieh et al., 2011*). Elevated levels of ecological plasticity among key ecosystem service providers such as *E. balteatus* are promising in view of the recent declines in global insect diversity and abundance (*Powney et al., 2019*; *Wotton et al., 2019*; *Sánchez-Bayo and Wyckhuys, 2020*). Nevertheless, our population genetics study encounters several shortcomings: (1) *E. balteatus* populations were primarily sampled in northern China and may not be representative of entire eastern Asia; (2) while conventional molecular markers proved to be effective for *E. balteatus* genetic analyses, NGS can yield a more in-depth characterization of genetic structure and evolutionary history.

Upon flower visitation, pollen is regularly deposited on insects' bodies and carried during flight (*Ahmed et al., 2009*). Pollen from geographically confined plant species can thus help to track insect dispersal and refine seasonal migration patterns (e.g., *Suchan et al., 2019*; *Guo et al., 2022*). Indeed, in our study, pollen grains from plants endemic to central or southern China (e.g., *Citrus* spp.) were routinely associated with spring–summer migrants. Meanwhile, pollen from *C. zawadskii* (i.e., restricted to Northeast China) was present in autumn migrants. As several *E. balteatus* individuals transported pollen grains over long-distance flights, they likely assume a prime role in transcontinental pollination and (insect-mediated) gene flow between geographically disjunct plant populations. Moreover, our work also identified the main (seasonal) foraging resources for *E. balteatus*. For example, spring–summer migrants relied extensively upon *T. mongolicum*, *A. altissima*, *A. fruticosa*, or *C. giganteum*, while *C. zawadskii* and *A. trifida* were frequently observed in autumn migrants. As these species may contribute to *E. balteatus* population build-up, these plant resources deserve further attention in nutritional ecology studies.

Molecular gut content analysis allows to decipher the plant hosts of insect herbivores and to unravel trophic interactions (*Avanesyan et al., 2021*). This technique is also increasingly used to uncover other trophic relationships, for example, predation or parasitism (e.g., *Symondson, 2002*; *Staudacher et al., 2011*; *Wang et al., 2017*). In this study, we successfully developed a DNA-based method to detect ingested plant DNA from *E. balteatus* guts. For a (comparatively small) set of plants, this method permitted detecting DNA in larval guts for up to 9-day postfeeding and comparing (plant- or herbivore-specific) consumption rates through qPCR. With the advent of the NGS approaches, molecular gut content analysis has become an exceptionally powerful tool – allowing high-throughput analysis of samples to simultaneously reveal multiple trophic linkages (*Pompanon et al., 2012*). Hence, by pairing gut content analysis with HTS, we detected 320 genera plant species from a total of 181 light-trapped *E. balteatus* on BH – the number of species is about ninefold greater than the 42 plant genera that were identified from 1014 BH adult migrants through pollen analysis. Drawing upon this sensitive approach, we registered links to 1012 plant species from 39 orders – far exceeding the number of host plants identified in previous studies (*Lucas et al., 2018a*; *Lucas et al., 2018b*). A wide spectrum of plants is thus visited by *E. balteatus* – comprising herbaceous plants, trees, and several cultivated crops. Hence, molecular gut content analysis presents proven advantages to decipher feeding relationships or map flower visitation networks. This method is less time consuming than palynological assays, which require the removal of pollen grains. Overall, our work offers a highly effective, sensitive approach to study the trophic interactions (and related coevolutionary processes) between hoverflies and their host plants. Future work can use these new methods to assess flower visitation networks, define the coevolutionary processes between hoverflies and their host plants, and to ultimately design strategies to bolster hoverflies' ecosystem service delivery. Aside from confirming the role of *E. balteatus* as a key pollinator of wildflowers (*Rader et al., 2020*; *Doyle et al., 2020*), gut content analysis can inform the design of habitat management and ecological engineering schemes to conserve this beneficial species in agricultural landscapes (*Landis et al., 2000*).

In summary, hoverflies are prime ecosystem service providers, which not only act as the second most important pollinators after bees but equally contribute to natural biological control of a broad suite of sap-feeding crop pests. Given the countless hoverfly individuals that annually disperse within the East Asia monsoon climatic zone, the social–ecological implications of *E. balteatus* ecosystem service delivery are non-negligible, that is, contributing to crop health and fruit yield (*Paschke et al., 2002*), while benefiting co-occurring (insect, vertebrate) pollinators, insectivores, and seed feeders. Aside from providing unique insights into *E. balteatus* migration ecology, our work constitutes a foundation for myriad follow-up experiments. For example, as several of the identified plant hosts likely mediate *E. balteatus* population build-up, further work is warranted to clarify their impacts on hoverfly nutritional ecology (*Pinheiro et al., 2014*) or biological control (*Batuecas et al., 2021*). Our findings can further aid the design of ecological engineering schemes to conserve *E. balteatus* in varying landscape contexts and to bolster its pollination and biological control services (*Landis et al., 2000*; *Gurr et al., 2016*). Lastly, NGS-based gut content analyses identification schemes – as the ones developed in our assays – could readily be built into biodiversity monitoring programs (*Reboud et al., 2022*).

## Materials and methods
### Light-trapping and field surveys
During 2003–2018, light trapping was conducted every night from April until October at a field station of the Chinese Academy of Agricultural Sciences (CAAS) at BH island (Shandong, China; 38°24′N; 120°55′E). With a size of approx. 2.5 km², BH is located in the center of China's Bohai Strait – an important migration pathway for multiple insect species which originate from the agricultural regions of mainland China at min. 40–60 km distances (*Feng et al., 2003*). High-altitude migrants of various insect species were collected using a vertical-pointing searchlight trap (model DK.Z.J1000B/t, 65.2 cm diam., 70.6 cm high, and ~30° spread angle; Shanghai Yaming Lighting Co., Ltd, Shanghai, China) equipped with a 1 000 W metal halide-lamp (model JLZ1000BT, Shanghai Yaming Lighting Co. Ltd, Shanghai, China) (*Feng et al., 2009*). This light-trap was mounted on a platform ~8 m above sea level. Except for events of heavy rain or power outage, the searchlight was switched on at a daily basis from sunset until sunrise. Trapped insects were collected into a 60-mesh nylon net bag – which was positioned under the trap and manually replaced every 2 hr throughout the night. Every day, *E.*

*balteatus* specimens were separated, counted, and a subset of trapped individuals was individually stored at −20°C in Eppendorf tubes for further analysis. Aside from a few pine trees, grasses, and gramineous weeds, there is no arable land on BH. Yet, to rule out the possibility that trapped *E. balteatus* individuals originated on BH itself, intensive field surveys were carried out throughout the monitoring period. Trapping data were used to describe *E. balteatus* population dynamics and to infer its migration patterns.

## Backward trajectory analysis

Backward trajectory analysis is widely used for inferring the movement patterns and population sources of migratory organisms (e.g., *Stefanescu et al., 2007*; *Huestis et al., 2019*; *Sun et al., 2021*). In this study, we used trajectory analysis to identify the possible origin of *E. balteatus* migrants that were trapped on BH island during spring–summer (April to July) and autumn (August to October). First, for trajectory simulation, we arbitrarily selected dates during 2003–2018 in which more than 40 *E. balteatus* individuals were caught (i.e., 'mass migration events'). A total of 76 'mass migration events' were thus identified, including 42 and 34 in the spring and autumn migration period, respectively (*Supplementary file 1a*). For each of those dates, meteorological data at a 1 × 1° resolution were extracted – through the Global Data Assimilation System (GDAS) – from the National Oceanic and Atmospheric Administration (NOAA) Air Resources Laboratory.

While *E. balteatus* primarily engages in diurnal flight, the BH searchlight trap was operated from sunset until sunrise of the subsequent day. Considering a max. 12-hr flight duration, each hour from 10:00 to 22:00 hr was set as a take-off time to simulate BH-bound immigration. When running the NOAA Hybrid Single Particle Lagrangian Integrated Trajectory (HYSPLIT) simulation model in MeteoInfo software (version 1.3.3) (*Wang, 2014*; *Stein et al., 2015*), the BH light-trap location was set as the end location. Based upon *E. balteatus* radar recordings (*Wotton et al., 2019*), simulations were performed for five flight heights, that is, 150, 300, 500, 800, and 1000 m above sea level. The *E. balteatus* flight speed was set identical to the wind speed, while flight duration was set to 12 hr. Trajectory analyses thus yielded migration endpoints, that is, take-off locations and potential source areas. Using ArcGIS software, we equally calculated the percentage of trajectories from a given region while omitting endpoints that fell into large water bodies.

## Hydrogen isotope analysis

To track animal migration movements, naturally occurring stable isotopes of various elements (e.g., H, C, and N) are efficiently used as endogenous markers. For migratory insects including hoverflies, the hydrogen isotope deuterium ($\delta D$) has been successfully applied (e.g., *Wassenaar and Hobson, 1998*; *Raymond et al., 2014*). In our study, we used $\delta D$ isoscape ratios to pinpoint the geographical origin of migratory *E. balteatus*.

### Hoverfly sampling

A total of 869 *E. balteatus* adults were collected. These included 286 individuals captured in the BH searchlight trap between April and October 2014–2018, and 583 individuals obtained through sweep-net sampling during April to October 2017–2018 at different sites across China. Sites were located in the Southwestern Region (SW; 1 site), Middle-Lower Yangtze Plain (YzP; 4), Northern Region (NP; 6), and Northeastern Region (NE; 3) (*Supplementary file 1b*). Each sample was preserved at −20°C and kept at the Institute for Plant Protection, CAAS in Beijing (China) until laboratory processing.

### Stable isotope recordings

As wing tissue is not part of the active metabolism after adult eclosion, it is generally used for isotope analyses (*Wassenaar and Hobson, 1998*). Accordingly, the wings of all *E. balteatus* specimens were removed with dissection scissors and subsequently sent to the Stable Isotope Mass Spectrometry Facility, Chinese Academy of Forestry (Beijing, China) for hydrogen isotope ($\delta D$) measurements as per *Zeng et al., 2020*. In brief, syrphid wings were cleaned with a methanol–chloroform solution (1:2) and air-dried overnight. Next, the hydrogen isotope ratio ($^2H{:}^1H$) of the combusted wings was measured using a Flash EA 1112 HT Elemental Analyzer (Thermo Fisher Scientific, Inc, USA) and Isotope Ratio and Mass Spectrometer (Delta V Advantage IRMS, Thermo Fisher Scientific, Inc, USA). Calculations

were done using the formula $\delta^2H‰ = (R_{sample}/R_{standard} - 1) \times 1000$ in which $R$ is the abundance ratio of heavy isotope to light isotope, namely $^2H/^1H$. The laboratory error was estimated to be ±2 ‰. Results are expressed in typical delta ($\delta D$) notation, in units of per mil (‰), and the relative standard of $\delta^2H$ was the Vienna Standard Mean Ocean Water (VSMOW).

## Population genetics studies

Population genetics studies are routinely used to assess adaptive capacity of different organisms including insects; the resulting data equally reveal migratory movements within an ecoevolutionary perspective (*Kim and Sappington, 2013*). In this study, we described *E. balteatus* genetic diversity and population structure using one mitochondrial gene and two nuclear genes.

### Specimen collections and DNA preparation

From April to October 2017–2018, a total of 670 *E. balteatus* adults were collected. These included 133 long-distance migrants that were caught in BH searchlight traps, and 537 individuals collected using sweep-net sampling at 16 sites (*Supplementary file 1b*). As above, sampling sites for the field-collected individuals were located within five geographical regions: Southwestern Region (SW; 1), Middle-Lower Yangtze Plain (YzP; 4), Northern Region (NP; 6), Northeastern Region (NE; 4), and Northwestern Region (NW; 1), covering three climatic regions, that is, mid-temperate zone, warm temperate zone, and subtropical zone. All samples were preserved at −20°C and kept at the Plant Protection Institute (CAAS, Beijing) until further processing. From each specimen, the total genomic DNA was isolated and extracted using a DNeasy Blood and Tissue Kit (Qiagen, Hilden, Germany), following the manufacturer's instructions. The extracted DNA was resuspended in 80 μl distilled water, and either used immediately or stored at −20°C for subsequent PCR analysis.

### PCR amplification and sequencing

For population genetic analyses, one partial mitochondrial (mt) gene [cytochrome b (*CYTB*)] and two nuclear genes [18S rRNA, 28S rRNA] were chosen as molecular markers. Primers were used as previously described (*Simmons and Weller, 2001*; *Mengual et al., 2015*), and synthesized by Sangon Biotech Co., Ltd (Shanghai, China). All fragment amplification was performed in a 25 μl PCR volume, using 2× GoldStar Master Mix (CWBIO, Beijing, China). Thermocycling conditions were 10 min at 95°C; followed by 35 cycles of 1 min at 95°C, 1 min at 55°C (COI) or 56°C (Cytb), and 45 s at 72°C, and a final extension of 72°C for 10 min. After verification through 2% gel electrophoresis, the resulting PCR products with the correct target size were sent to Beijing Genomics Institute (BGI) Co., Ltd (Beijing, China) for sequencing in both directions.

### Genetic diversity and population genetic structure

Sequencing results were manually edited, checked, and assembled with Chromas 2.31 (Technelysium, Helensvale, Australia) and Seqman within the Lasergene suite version 7.1.2 (DNASTAR, Inc, USA). The corrected nucleotide sequences were then aligned using the ClustalW algorithm implemented in MEGA 6.0 with default parameters (*Tamura et al., 2013*). To assess *E. balteatus* genetic diversity, the following parameters were calculated for the entire dataset and each individual population using DNASP 6.0 (*Rozas et al., 2017*): number of polymorphic sites (*S*), number of haplotypes (*h*), haplotype diversity (Hd), nucleotide diversity (Pi), and average number of nucleotide differences (*K*). Geographical distribution profiles, phylogenetic trees and haplotype networks were also used to visualize the genetic linkages of different subpopulations. Phylogenetic trees were constructed using the maximum likelihood method with 1000 bootstrap replicates in MEGA 6.0 (*Tamura et al., 2013*), while haplotype networks were built in Network 4.6 with the median-joining algorithm (*Bandelt et al., 1999*). To further evaluate the degree of population differentiation, an analysis of molecular variance and pairwise population differentiation ($F_{ST}$) were carried out using Arlequin 3.059 with 10,000 random permutations (*Excoffier and Lischer, 2010*). Genetic distances between subpopulations were calculated with a coalescent-based approach using the Bayesian search strategy implemented in MIGRATE-N v. 3.2.1 (*Beerli, 2006*). To infer demographic history, we used two neutrality tests, that is, *Tajima, 1989* and *Fu, 1997*, and mismatch distribution analysis (*Rogers and Harpending, 1992*). Tajima's *D* and Fu's $F_S$ are expected to be nearly zero in an effective population of constant size, with negative or positive values indicative of a respective population expansion or recent bottleneck (*Fu,*

*1997*). Populations at demographic equilibrium exhibit a multimodal mismatched distribution, while unimodal patterns reflect recent demographic or area expansions (*Rogers and Harpending, 1992*). All analyses were conducted using Arlequin 3.5 (*Excoffier and Lischer, 2010*).

## Pollen grain analysis

Pollen grains carried by insects are routinely used to assess feeding history or movement patterns (e.g., *Jones and Jones, 2001*). In this study, we used pollen grains that adhered to the body of migrating *E. balteatus* adults and integrated morphologically based approaches through scanning electron microscopy (SEM) and molecular tactics based upon DNA barcoding. This twin method has been previously employed in our laboratory (e.g., *Liu et al., 2016*).

### Sample collection, pollen preparation, and SEM examination

Over 2014–2018, several subsamples of 20 migratory *E. balteatus* (or all individuals if the total capture size was below 20) were randomly taken from the BH searchlight-trap sample. As such, a total of 1014 individuals were obtained (*Supplementary file 1c*). First, all collected samples were examined at ×200 magnification using a stereomicroscope (Olympus SZX16, Pittsburgh, PA, USA). Next, suspected pollen grains were gently removed from the insect body and mounted on aluminum stubs with double-sided sticky tape. Next, pollen samples were sputter-coated with gold, and visualized with a Hitachi S-8010 cold field emission scanning electron microscope (Hitachi, Tokyo, Japan) at the Electronic Microscopy Centre of the Institute of Food Science and Technology (CAAS, Beijing, China), or a Zeiss Field Emission scanning electron microscope (Merlin, Zeiss, Germany) at the National Center for Electron Microscopy and School of Materials Science and Engineering, Tsinghua University, Beijing.

### Molecular analysis of single pollen grains

Genomic DNA was extracted from single pollen grains using protocols adapted from *Chen et al., 2008*. In brief: pollen grains were transferred to individual PCR tubes that contained 5 µl of lysis solution (0.1 M NaOH plus 2% Tween-20), and incubated for 17 min 30 s at 95°C in a thermocycler (GeneAmp PCR System 9700, Applied Biosystems, Foster City, CA, USA). For each lysis solution, 5 µl Tris–Ethylene Diamine Tetraacetic Acid (EDTA) (TE) buffer was added and the resulting solution was used as a template for subsequent PCR amplifications. To improve species-level identification, four DNA barcoding loci for plants were used simultaneously, that is, two mitochondrial spacer elements ITS1 and ITS2, and chloroplast *rbcL* (*Fay et al., 1997*; *Fazekas et al., 2008*; *Cheng et al., 2016*). All partial regions were separately amplified using DreamTaq DNA polymerase (Thermo Fisher Scientific, Waltham, MA) with the following conditions: an initial denaturation step (95°C for 3 min), followed by 38 cycles at 95°C for 1 min, 55°C for 30 s, 72°C for 1 min, and a final extension of 10 min at 72°C. The resulting PCR products were gel purified with a Gel Extraction Kit (TransGen, Beijing, China) and ligated directly into the pClone007 Vector (Tsingke, Beijing, China) or pEasy-T3 vector (TransGen, Beijing, China). Positive clones were randomly selected and sequenced with M13 primers using Sanger sequencing at Sangon Biotech Co., Ltd (Shanghai, China) or Tsingke Biotechnology Co., Ltd (Beijing, China).

### Pollen and plant host identification

Each pollen grain was identified based upon its molecular and morphological characteristics, and geographic distribution. First, using the online BLASTn search program, genetic sequences were compared with those at the National Center for Biotechnology Information (NCBI) database. If the sequence top bit score matched with a single species, multiple species within a given genus or multiple genera within a given family, then the sequence was designated to the respective species, genus, or family. Sequences that aligned with multiple families were termed to be 'unidentifiable' (*Hawkins et al., 2015*). As such, several sequence taxa were assigned to the rank of genus or family. Separate analyses were performed for the four tested markers and results were combined to identify a given pollen species. Identifications based upon molecular data were further complemented by morphological characterization, using published SEM images of pollen grains of Chinese flora (*Ma et al., 1999*; *Li et al., 2010*) or online search engines and palynological databases (https://www.paldat.org/). Finally, species-level identifications were checked against the Flora of China Species Library (https://species.

sciencereading.cn) and Plant Science Data Center (https://www.plantplus.cn/cn), to determine the presence of plant hosts within the broader study area.

## Molecular gut content analysis

Molecular gut content analysis has been successfully used to clarify trophic relationships such as herbivory, predation, or parasitism (e.g., *Staudacher et al., 2011*). In this study, laboratory assays were conducted to identify the plant hosts of pollinivorous *E. balteatus*. Specifically, experimental feeding assays were run to quantitatively assess identification accuracy and detectability half-lives for plant DNA through diagnostic or quantitative real-time PCR (qPCR).

### Insect rearing and pollen collection

For this experiment, *E. balteatus* individuals were sourced from a laboratory population. A laboratory colony was established from individuals sourced in wheat fields at the CAAS Experimental Station in LangFang (LF; Hebei, China; 39.53°N, 116.70°E) during May 2017, and maintained at 25 ± 1°C, 65% ± 5% relative humidity (RH), and a 16:8 L:D photoperiod. Hatched *E. balteatus* larvae were fed on *Megoura japonica* aphids, and adult flies were fed 10% (wt/vol) honey solution and pollen (*Li et al., 2021*). As experimental food in feeding assays, pollen was provided of three preferred host plants: *Cannabis sativa*, *Humulus scandens*, and *Helianthus annuus*. This pollen was directly collected from flowering plants near LF experimental fields during autumn of 2018. After collection, pollen of a given plant species was stored separately in a sterile recipient and kept refrigerated at 4°C.

### Feeding assay

Newly emerged *E. balteatus* adults were held individually in a disposable plastic cup (30 × 30 × 30 cm), provided with a 10% (wt/wt) sucrose solution for 1 day, and subsequently starved for 12 h. Access to water was ensured through a saturated cotton wick. Next, healthy and active hoverfly adults were randomly assigned to either of three diets. More specifically, a given *E. balteatus* adult was individualized in (10 cm diam., 2.6 cm high) plastic Petri dishes containing either of target pollen species, that is, *C. sativa*, or *H. scandens* or *H. annuus*. Assays were run at room temperature. Over the span of 1 hr, individuals were observed every 5 min to confirm feeding. Next, those individuals that had fed at least 15 min were selected and transferred to a plastic cup. In this cup, *E. balteatus* adults were given access to a moistened cotton ball (water only) and kept for 0, 2, 4, 8, 16, 24,32, 48, and 72 hr. For each treatment, a minimum of eight samples (replicates) were obtained and individual adults were freeze-killed immediately at each time point. Next, each hoverfly adult was individualized within 1.5 ml tubes and stored at −20°C for subsequent molecular assay. Some adults were not allowed to feed and were freeze-killed at 0 hr – thus serving as negative controls.

### DNA extraction

Total DNA was extracted from dissected abdomens with the DNeasy Blood and Tissue Kit (Qiagen, Hilden, Germany), according to the manufacturer's recommendations. DNA extracts were normalized to 20 ng/μl using Qubit fluorimeter quantification (Invitrogen), and either used directly for PCR amplification or stored at −20°C until further processing. To avoid plant DNA contamination from the hoverfly's outer body surface, a bleaching method was adapted from earlier studies and used prior to DNA extraction (*Wallinger et al., 2013*). In brief: each individual was washed with 1.5% NaOCl (Beijing Chemical works, Beijing, China) for 10 s and then rinsed with molecular analysis-grade water. Preliminary trials showed that this removed plant DNA contamination from the body surface without destroying the ingested DNA in *E. balteatus* guts. To eliminate any further environmental contamination, all working spaces and equipment were regularly cleaned with 70% ethanol.

### PCR amplification and sequencing

To identify pollen loads or host plants from insects, ITS2 offers a high success rate, enables subsequent amplicon sequencing, and counts with a high number of reference sequences among plant DNA barcodes (*Han et al., 2013*; *Pang et al., 2013*). Universal ITS2-targeted primers (*Cheng et al., 2016*) were thus used to specifically detect plant DNA in the feeding assay. PCR assays, TA cloning, and sequencing were performed as above. The resultant sequences were identified using BLASTn,

and corrected plasmids containing the cloned fragments were used to construct a quantification standard curve.

### Quantitative real-time PCR

For each of the three plant species, we quantified the amount of DNA present in a given *E. balteatus* individual at different times with the respective primer pair through qPCR. Species-specific primers and probes were designed from sequences using the above universal ITS2-targeted primers and synthesized by BGI (Beijing, China) (*Supplementary file 1d*). qPCRs were performed with the TaqMan method in 20 µl reaction agents composed of 10 µl of 2× QuantiTect Probe PCR Master Mix (Qiagen, Hilden, Germany), 0.5 µM of each primer, 0.2 µM probe, and 1 µl of template DNA, using a 7500 Fast Real-time PCR System (Applied Biosystems). Thermocycling conditions were as follows: 95°C for 2 min, followed by 40 cycles of 94°C for 10 s, 60°C for 10 s. To avoid technical errors, all qPCR reactions were replicated three times. To quantify the amount of DNA of *C. sativa*, *H. scandens*, and *H. annuus*, the respective standard curve equations were used: $y = -0.3239x + 12.313$ ($R^2 = 0.9975$), $y = -0.3014x + 12.372$ ($R^2 = 0.9994$), and $y = -0.3092x + 12.505$ ($R^2 = 0.9993$) (*Figure 6—figure supplement 1*). In the above equations, $y$ equals to the logarithm of plasmid copy number to base 10, while $x$ = Ct value.

## DNA metabarcoding of gut contents

Metabarcoding has revolutionized species identification and biomonitoring using environmental DNA, offering bright prospects for dietary analyses (*Pompanon et al., 2012*). In this study, we combined PCR-based gut content analysis with HTS of field-collected individuals. The host plant identification protocol could thus be validated under 'real-world' conditions and further insights can be gained into *E. balteatus* adult food choice.

### Sampling and DNA extraction

Two sets of adult *E. balteatus* specimens were collected: 180 long-distance migrants captured in the BH searchlight traps from April to October 2015–2018, and 434 individuals collected through sweep-net sampling at 19 sites from April to October 2017–2018. Field-caught populations originated from five geographical regions of China (*Supplementary file 1b*). DNA from field-caught samples was individually extracted using the DNeasy Blood and Tissue Kit (Qiagen, Hilden, Germany), and subject to experimental procedures as described above.

### MiSeq sequencing of ITS2 barcode gene amplicons

A two-step laboratory protocol was followed: the ITS2 fragment was first amplified using the barcoded universal primers to generate mixed amplicons. Next, amplicons were sequenced using HTS for comparison against known barcode references. In the first step, we used PCR to uniquely index each sample using the modified universal primers that were tagged with a sample-specific eight-mer oligo-nucleotide tag at the 5'-end. In total, 90 sets of index primers were used to amplify the ITS2 region. This was done to ensure that multiple samples could be processed simultaneously into a single sequencing run and be subsequently separated via bioinformatics processing. Each sample was processed in three independent PCRs to avoid reaction-specific biases. Each of the replicate PCRs consisted of 2 µl of the DNA template, 12.5 µl of 2× GoldStar Master Mix (CWBIO, Beijing, China), and 0.5 µM of amplicon primers in a 25 µl reaction volume with the following PCR program: 95°C for 10 min, 38 cycles of denaturation at 95°C for 1 min, annealing at 49°C for 40 s and elongation at 72°C for 40 s, and a final extension step at 72°C for 10 min. After reaction, the replicate PCRs for each sample were combined and gel-purified using the MinElute Gel Extraction Kit (QIAGEN, Hilden, Germany) as per the manufacturer's instructions and quantified with a NanoDrop ND-2000 UV-vis spectrophotometer (Nano-Drop Technologies, Wilmington, DE). Next, purified amplicons with different tags were pooled in an equal concentration to make a composite DNA sample and preceded for HTS sequencing. A negative control with no DNA template was run in parallel. All pooled amplicons (up to 90 specimens each) were sent to the GENEWIZ, Inc (Suzhou, China) for final amplicon library construction and Illumina HTS. Briefly, for each mixed-amplicon, the indexed Illumina-compatible libraries were constructed using VAHTS Universal DNA Library Prep Kit for IlluminaV3 (Vazyme Biotech Co., Nanjing, China) with

standard protocols, and subsequently sequenced on the Illumina MiSeq platform (Illumina, San Diego, CA, USA) using a 2 × 300 bp paired-end configuration, as per the manufacturer's protocols.

## Data analysis

Paired-end reads were assigned to samples according to the unique 8 bp barcode of each sample and truncated by cutting off the barcode and primer sequence, reads containing ploy-N, and low-quality reads. The trimmed forward and reverse reads were joined based on overlapping regions within paired-end reads. Next, chimeric sequences were removed after being identified by comparing merged sequences with the reference RDP Gold database using UCHIME algorithm. The remaining high-quality reads were then clustered into OTUs at a 99% sequence similarity level, using VSEARCH1.9.6. Representative OTU sequences were taxonomically annotated with BLASTn searches against the NCBI database (accessed 02/2021) using the remote command line interface (*Camacho et al., 2009*). For each sequence, the top 50 matches were written to a tsv file, but matches were only included in downstream analyses once they met the following criteria: (1) min. 90% of the query sequence is present in the BLASTn-generated subject sequence (sequence coverage), (2) at least 95% sequence similarity (identity), (3) an *e*-value below 0.001, and (4) a subject sequence derived from ITS2 of a land plant (*Prosser and Hebert, 2017*). All samples that yielded a sequence that met these thresholds from one or both PCR replicates were considered as host identifications. Annotation results were analyzed following *Hawkins et al., 2015*. Furthermore, all BLAST results were verified using expert knowledge, integrating an understanding of local habitats, species distribution, and rarity to refine plant host identifications. For each dietary item, occurrence frequency was calculated as the number of samples in which the item was present divided by the total number of samples. Also, the relative read abundance was calculated by dividing the number of reads of each dietary item (and individual) by the total number of reads in the sample.

## Statistical analysis

Differences in *E. balteatus* trap capture rate were analyzed using a zero-inflated generalized linear model (*Brooks et al., 2017*). To compare counts among years, post hoc tests were run using the emmeans package (*Searle et al., 2012*). Wilcoxon rank-sum test was used to compare differences in the δD values of migratory *E. balteatus* wings sampled at different times. A chi-square test was used to compare the differences in the frequency of pollen deposits on *E. balteatus* during different migration phases and the characteristics of pollen source plants. To compare δD and alpha diversity values between groups, we used One-way analysis of variance followed by Tukey's test for multiple comparisons. Venn and upset diagrams were drawn to represent the interactions among host plant communities of different groups using the omicshare cloud tool under default instructions (http://www.omicshare.com/). PCA and analysis of similarities (ANOSIM) were used to evaluate differences in *E. balteatus* host plant abundance between specific migration phases. To identify features characteristic of certain groups, we built a random forest machine learning model based on genus abundance data from 180 *E. balteatus* migrants captured on BH. All statistical analyses were performed in R version 4.0.3 (*R Core Team, 2020*).

## Acknowledgements

We would like to express our thanks to all of the people who kindly helped us during wild samples collecting, especially Prof. Peng Wan (Institute of Plant Protection and Soil Science, Hubei Academy of Agricultural Sciences), Prof. Jian Liu (College of Agriculture, Northeast Agricultural University), Prof. Honghua Su (College of Horticulture and Plant Protection, Yangzhou University), Prof. Yutao Xiao (Agricultural Genomics Institute at Shenzhen, Chinese Academy of Agricultural Sciences), Prof. Xingya Wang (College of Plant Protection, Shenyang Agricultural University), Dr. Lili Wang (Yantai Academy of Agricultural Sciences), and Ms. Aili Li (Xinxiang Experimental Station of Chinese Academy of Agricultural Sciences). We also thank Prof. Zhiheng Wang (Peking University) for providing the atlas of plants in China.

## Additional information

### Funding

| Funder | Grant reference number | Author |
|---|---|---|
| The Laboratory of Lingnan Modern Agriculture Project | NT2021003 | Kongming Wu |
| The Agricultural Science and Technology Innovation Program Cooperation and Innovation Mission | CAAS-XTCX2018022 | Kongming Wu |

The funders had no role in study design, data collection, and interpretation, or the decision to submit the work for publication.

### Author contributions

Huiru Jia, Investigation, Methodology, Writing - original draft; Yongqiang Liu, Hui Li, Chaoxing Hu, Investigation, Methodology; Xiaokang Li, Yunfei Pan, Xianyong Zhou, Formal analysis, Investigation, Methodology; Kris AG Wyckhuys, Writing - review and editing; Kongming Wu, Formal analysis, Funding acquisition, Project administration, Supervision, Writing - review and editing

### Author ORCIDs

Kongming Wu (iD) http://orcid.org/0000-0003-3555-4292

### Decision letter and Author response

Decision letter https://doi.org/10.7554/eLife.76230.sa1
Author response https://doi.org/10.7554/eLife.76230.sa2

## Additional files

### Supplementary files

• Supplementary file 1. Data supporting the findings of this study. (a) Mass migration events of *E. balteatus* across the Bohai Strait observed by the searchlight trapping on BH Island during 2003–2018. (b) Collection information for sample in the study. (c) Pollen carrying rate of the migratory *E. balteatus* hoverflies across Bohai Sea during 2014–2018. (d) Quantitative PCR (qPCR) primers and conditions used in this study. (e) Key parameters within *E. balteatus* searchlight trapping on BH Island during 2003–2018. (f) The percentage of the total trajectories that ended in each region. (g) Results of analysis of molecular variance (AMOVA) test in different populations and regions of *E.balteatus* based on Cytb and18S-28S rRNA gene. (h) Comparative assessment of the degree of taxonomic identification obtained through either molecular or morphology-based approaches, for 46 different types of pollen grains dislocated from *E. balteatus* long-distance migrants collected on Beihuang Island (Bohai Sea, northeastern China). For each type of pollen grain, the highest level of taxonomic identification is indicated and contrasted between molecular and morphology-based approaches.

• Supplementary file 2. List of sequence information of each sample in molecular gut analysis.

• Supplementary file 3. Absolute abundance of host plants for each sample based on molecular gut analysis.

• Transparent reporting form

### Data availability

The raw MiSeq data from DNA metabarcoding of gut contents have been deposited at NCBI Sequence Read Archive (SRA) under BioProject PRJNA816296. All data supporting the findings of this study are available within the Article, the Extended Data, and the Supplementary Information files.

The following dataset was generated:

| Author(s) | Year | Dataset title | Dataset URL | Database and Identifier |
|---|---|---|---|---|
| Wu K | 2022 | Species identification of plant tissues from the gut of Episyrphus balteatus | https://www.ncbi.nlm.nih.gov/bioproject/?term=PRJNA816296 | NCBI BioProject, PRJNA816296 |

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
