## [Editor Report]

Hoverflies are a group of insects which provide crucial ecosystem services such as pollination and crop protection. Their migratory behavior in western countries is well characterized, but in eastern Asia, the annual summer monsoon provides a 'highway' of favorable winds for the airborne transport of migratory organisms, and the migration of hoverflies in this large region has not been well studied. This study addresses hoverfly migration in East Asia and its consequences using a variety of suitable methods and will be of great interest to insect migration biologists and pollination ecologists.

---

## [Decision Letter]

**Decision letter after peer review:**

Thank you for submitting your article "Windborne migration amplifies insect-mediated pollination services" for consideration by *eLife*. Your article has been reviewed by 3 peer reviewers, including Youngsung Joo as Reviewing Editor and Reviewer #1, and the evaluation has been overseen by Meredith Schuman as the Senior Editor. The following individuals involved in review of your submission have agreed to reveal their identity: Jason Chapman (Reviewer #2); Gao Hu (Reviewer #3).

Essential revisions:

Please respond only to these essential revisions, which are determined from the consultative review process, in your response letter and your revision. In doing so you may refer to relevant comments from the individual reviews.

1) None of the reviewers are convinced that hoverflies migrate at night on purpose, and suggest that other factors may contribute to their behavior at night. According to Reviewer 2 (major comment 2) and Reviewer 3 (comment ii), the nocturnal migration of the hoverfly can possibly be explained by the geographical location of the trap (an island): specifically, the hoverflies likely start to migrate during the day but, at the end of the day, find themselves over water and so must keep flying until they reach land (the island). Please address this concern in your revision.

2) As a result of the species' diurnal activity, a flight lasting up to 24 hours is highly unlikely. A flight lasting no more than 12 hours is a more likely maximum duration, with flights typically lasting no longer than eight hours (please see Reviewer 2 major comment 3). Please address this concern and revise your trajectory analyses accordingly.

3) Many interesting outcomes are described, but insufficient explanations are provided (please see detailed comments from all three reviewers). The data should be discussed in more detail. A large part of the present manuscript was devoted to the methods section, despite the fact that the manuscript was already quite long. The main text of *eLife* consists of about 5000 characters, excluding the methods part, so there is ample space for a detailed description of the results and their discussion. All three reviewers provide suggestions for how to focus the text, and for which aspects need more presentation in the manuscript. Please substantially revise the manuscript along these lines and explain these revisions in your response letter. In doing so it is not necessary to follow the specific recommendations of all reviewers, but you must convincingly address the general concern that the results are not sufficiently presented and discussed.

4) Along the same lines, the figures, especially Figures 4-8, should be revised substantially in terms of data presentation. These present too much information that cannot be distilled into solid take-home messages, and many of the subfigures are auxiliary. Furthermore, the data from different regions are not seasonally aligned, so they cannot be compared with the data from BH. Please substantially revise the presentation of the data in the figures and explain in your response letter how this has been revised to make each figure coherent, and support rigorous conclusions. Please see suggestions especially from Reviewers 1 and 3. Again, in doing so, you do not need to respond to each individual reviewer comment.

*Reviewer #1 (Recommendations for the authors):*

Format

The main text of the article should usually be structured and ordered as follows: Introduction; Results; Discussion; Materials and methods (or Methods); Acknowledgements; References; Figures with the corresponding legend below each one; and Tables.

Figure 1.

Migration fluctuations were quite strong each year. Is there any explanation for this? Further, the authors suggest that this fluctuation is related to outbreaks of aphids, but no data or references are provided. Please provide further details.

Figure 2.

According to the literature, hydrogen isotope data should be interpreted in relation to elevation. Therefore, I believe it is necessary to present elevation information for each place and interpret it accordingly.

Figure 3.

The neutrality test indicates that PE and SH can be isolated from other populations. However, the authors did not interpret the details of the neutrality test. In addition, although the authors provided regional information, Figure 3D is still difficult to comprehend for those who are not Chinese. It would be easier to understand if each circle was positioned on a map of China. Lastly, I think Figures 3A and B should be included in the supporting information.

Figure 4.

It is well known that hoverflies are generalist pollinators. Therefore, the results showing that hoverflies carry various pollen grains are not considered significant. It would be useful to include a graph showing whether the seasonal migration pattern, which is the take-home message of this study, is well expressed in plants carried by migrants. Particularly, the author discussed that it is possible to estimate whether the hoverfly has passed through a specific ecological zone based on plants that only inhabit a particular geographical area. However, no data or references are provided for L501 – 506. Please provide further details.

Figure 5,6,7,8.

Since each region has a different sampling period, the data appear to be incomparable. For instance, the seasons surveyed in the CD region appear to be longer than those in other regions. I am not sure if this interpretation is correct since the vegetation structure can change according to the season. Therefore, I recommend focusing on the data from BH. It would be helpful if you could provide a correlation between the phenology and spatial distribution of migrants collected from BH (or CD?) as they change by season.

*Reviewer #2 (Recommendations for the authors):*

I have three major areas where I suggest the manuscript should be modified.

1. Firstly, I believe the manuscript is too short to adequately deal with all of the strands of the paper. There are four major components to the results: (1) the trajectory analyses; (2) the pollen analyses; (3) stable isotope analyses; (4) the population genetics analyses. Because of the length constraints of the manuscript, each of these sections is dealt with quite briefly, and the manuscript suffers because of this. For example, the stable isotope analyses and the genetic analyses are both rather complex topics that really need a lot more explanation to allow the reader to follow the results and the conclusions. My feeling is that it will be impossible to adequately cover all of the strands within a single paper. My preferred option would be for the genomic component of the paper to be separated and published as a stand-alone study, in which the authors could properly document their findings and the implications, which could then be examined by a specialist referee. The other strands of the paper I believe do fit together, but the extra space would allow each of the sections to be expanded upon and more fully described. As it stands, the paper is not publishable in my opinion, because each of the sections is too briefly documented.

2. Secondly, I do not agree that the data presented support the authors' contention that the study species routinely migrates at night. The light-trap catches from the island at first glance seem to support this. However, the most likely scenario is that this day-active species would initiate migration during the daytime, but if it finds itself over the sea as the sun sets, then of course it will continue to fly into the night because landing is impossible. Upon reaching the island/seeing the light, the hoverflies are caught, giving the impression of nocturnal activity. But if this light-trap had been on a mainland site, I strongly suspect that nocturnal catches would not occur. In my research group, we have carried out aerial sampling of migrant insects high above the ground on hundreds of days and nights, in many locations around the world – we have regularly caught hoverflies during daytime samples, but never caught one at night. I work closely with Karl Wotton and his research group, who has been studying hoverfly migration through high Alpine passes in the Pyrenees, catching insects both day and night. They have caught tens of thousands of hoverflies during daytime samples, and never caught them at night. I believe these field data strongly support the widely-held contention that hoverflies are diurnal migrants. Thus I do not agree with the statement made in the abstract of current paper: "Given the substantial night-time dispersal of E. balteatus, this species possibly adopts a 'dual' migration strategy." The authors should modify their text to reflect this view.

3. Following on from the previous point. The authors run the trajectory analyses backwards from the time of (nocturnal) light-trap catches, for 24 hours. This indicates that the authors are assuming the hoverflies started migration during the night, and continued non-stop for 24 hours. Neither of these contentions are at all likely. As stated above, the hoverflies would have taken off during daytime hours (probably from about mid-morning onwards, as air temperatures rise). And flight for as long as 24 hours seems highly unlikely given the diurnal activity of the species – 12 hours of non-stop flight seems like a more likely maximum duration, with flight typically being less than eight hours non-stop. The trajectory analyses really should be recalculated using more realistic flight parameters.

*Reviewer #3 (Recommendations for the authors):*

Figure 1A, the circular histogram of annual catches looks very strange, I think it should be better to show in simple histogram with cartesian coordinates.

Figure 2B could be removed into the Supplementary, and the sites in Figure 2A also don't need be labeled.

Figure 3 is very complicated, I couldn't catch the main idea of this figure. I am not expert on molecular or genetics studies. For me, I am very interesting on the geneflow, and I think Figure 3B and 3D are the most important. I really hope the paper can give more explanation, why the pattern from mitochondrial and nuclear was different? Why the general migration trend towards the Yangtze basin?

Figure 4 and 5 could be merged together. Firstly, I don't think it is necessary to show all pollen pictures of 46 species. Secondly, map in Figure 5A and 2A is very similar, is it necessary showed it again and again? Third, Figure 5B did not showed very important information and might be better to move it to the Supplementary.

Figure 6-8, only a very short paragraph described these three large figures. I think each figure should be simpler, and need more description and more explanation.

[Editors’ note: further revisions were suggested prior to acceptance, as described below.]

Thank you for resubmitting your work entitled "Windborne migration amplifies insect-mediated pollination services" for further consideration by *eLife*. Your revised article has been evaluated by Meredith Schuman (Senior Editor) and Youngsung Joo (Reviewing Editor).

The manuscript has been improved but there are some remaining issues that need to be addressed, as outlined below.

Essential revisions:

Please address these three points in your revision and response letter. The individual reviews are provided for your information and will give details to guide your response to the essential revisions, but you do not need to provide a point-by-point response to the reviews – only to the essential revisions.

1) Please elaborate on figures 8 and 9 in the results. Currently, even experienced insect migration researchers are still having some difficulty with these figures, so any extra clarification you can provide regarding the take-home message of these figures and their relevance to the story would be helpful. Alternatively, these figures could be moved to supplementary material and briefly summarized in the text.

2) Describe the geographical and biological background of Beihuang Island in the introduction, as the Materials and methods have been moved to the end of the paper.

*Reviewer #1 (Recommendations for the authors):*

I would like to make one more comment before the acceptance. It is necessary for the introduction to describe the geographical and biological background of Beihuang Island as the Materials and methods have been moved to the end of the paper.

*Reviewer #2 (Recommendations for the authors):*

The authors have done a fine job in implementing some of the changes that I suggested – most importantly, I was concerned about the long trajectory durations and the assumption that nocturnal flight was the norm, and so I am pleased that they have made important changes in the manuscript with respect to these points.

In general, the paper is written to a high standard throughout, and contains some extremely interesting data. The study represents an exceptionally detailed and thorough examination of an insect migration system, and is very worthy of publication.

I still feel (personal opinion) that perhaps too much material is presented, as some sections (particularly the explanation of Figures 8 and 9) are not explained clearly enough for me – but this is perhaps something for the editorial board to make a decision on.

---

## [Author Response]

Essential revisions:Please respond only to these essential revisions, which are determined from the consultative review process, in your response letter and your revision. In doing so you may refer to relevant comments from the individual reviews.1) None of the reviewers are convinced that hoverflies migrate at night on purpose, and suggest that other factors may contribute to their behavior at night. According to Reviewer 2 (major comment 2) and Reviewer 3 (comment ii), the nocturnal migration of the hoverfly can possibly be explained by the geographical location of the trap (an island): specifically, the hoverflies likely start to migrate during the day but, at the end of the day, find themselves over water and so must keep flying until they reach land (the island). Please address this concern in your revision.

We agree with the reviewers’ perspective and have now removed reference to a “dual migration strategy” from the Abstract section. Furthermore, we have expanded our discussion on this topic -according to reviewers’ comments- in lines 312-328.

2) As a result of the species' diurnal activity, a flight lasting up to 24 hours is highly unlikely. A flight lasting no more than 12 hours is a more likely maximum duration, with flights typically lasting no longer than eight hours (please see Reviewer 2 major comment 3). Please address this concern and revise your trajectory analyses accordingly.

We are grateful for these valuable comments. We have now revised our trajectory analyses and recalculated backward trajectories every hour from 10:00 to 22:00, by setting the max flight duration to 12 h.

3) Many interesting outcomes are described, but insufficient explanations are provided (please see detailed comments from all three reviewers). The data should be discussed in more detail. A large part of the present manuscript was devoted to the methods section, despite the fact that the manuscript was already quite long. The main text of eLife consists of about 5000 characters, excluding the methods part, so there is ample space for a detailed description of the results and their discussion. All three reviewers provide suggestions for how to focus the text, and for which aspects need more presentation in the manuscript. Please substantially revise the manuscript along these lines and explain these revisions in your response letter. In doing so it is not necessary to follow the specific recommendations of all reviewers, but you must convincingly address the general concern that the results are not sufficiently presented and discussed.

We wish to thank all reviewers for these comments. As suggested, we have now carefully addressed all concerns and adapted our manuscript accordingly. Aside from addressing the specific recommendations of individual reviewers, we have also attempted to improve the overall presentation (and discussion) of our key findings.

4) Along the same lines, the figures, especially Figures 4-8, should be revised substantially in terms of data presentation. These present too much information that cannot be distilled into solid take-home messages, and many of the subfigures are auxiliary. Furthermore, the data from different regions are not seasonally aligned, so they cannot be compared with the data from BH. Please substantially revise the presentation of the data in the figures and explain in your response letter how this has been revised to make each figure coherent, and support rigorous conclusions. Please see suggestions especially from Reviewers 1 and 3. Again, in doing so, you do not need to respond to each individual reviewer comment.

As suggested, we have reworked the Results -and specifically Figures 4-8. We have also strengthened the overall presentation of the data e.g., by also amending other figures. Below, we provide more detailed information on how specific elements of the Results section were revised.

Reviewer #1 (Recommendations for the authors):FormatThe main text of the article should usually be structured and ordered as follows: Introduction; Results; Discussion; Materials and methods (or Methods); Acknowledgements; References; Figures with the corresponding legend below each one; and Tables.

We have now changed our article structure as suggested.

Figure 1.Migration fluctuations were quite strong each year. Is there any explanation for this? Further, the authors suggest that this fluctuation is related to outbreaks of aphids, but no data or references are provided. Please provide further details.

In the initial draft of our manuscript, we discussed the (potential) determinants of the (strong) inter-annual fluctuations in migration patterns in the second paragraph of the Discussion section. However, in order to clarify this further to future readers, we now discuss this topic in further detail in lines 291-311.

Figure 2.According to the literature, hydrogen isotope data should be interpreted in relation to elevation. Therefore, I believe it is necessary to present elevation information for each place and interpret it accordingly.

We agree with reviewer #1 that elevation does affect the hydrogen isotope ratio. However, when sampling is carried out over a relative narrow elevational range, its impact on isotope ratio is rather small. As all our sampling was conducted at low elevation (<1000 masl), we have opted not to include elevational data in Figure 2A.

Figure 3.The neutrality test indicates that PE and SH can be isolated from other populations. However, the authors did not interpret the details of the neutrality test. In addition, although the authors provided regional information, Figure 3D is still difficult to comprehend for those who are not Chinese. It would be easier to understand if each circle was positioned on a map of China. Lastly, I think Figures 3A and B should be included in the supporting information.

We agree with the reviewer’s assessment. We have opted to maintain Figures 3B and D in the main body of the manuscript, while transferring Figures 3A and C to the Supplementary Information.

Figure 4.It is well known that hoverflies are generalist pollinators. Therefore, the results showing that hoverflies carry various pollen grains are not considered significant. It would be useful to include a graph showing whether the seasonal migration pattern, which is the take-home message of this study, is well expressed in plants carried by migrants. Particularly, the author discussed that it is possible to estimate whether the hoverfly has passed through a specific ecological zone based on plants that only inhabit a particular geographical area. However, no data or references are provided for L501 – 506. Please provide further details.

We are grateful to reviewer #1 for this excellent suggestion. As recommended, we have now added a graph showing the extent to which seasonal migration patterns are mirrored in hoverfly pollen loads. We equally extended our discussion on these findings and analysis in lines 300-500.

Figure 5,6,7,8.Since each region has a different sampling period, the data appear to be incomparable. For instance, the seasons surveyed in the CD region appear to be longer than those in other regions. I am not sure if this interpretation is correct since the vegetation structure can change according to the season. Therefore, I recommend focusing on the data from BH. It would be helpful if you could provide a correlation between the phenology and spatial distribution of migrants collected from BH (or CD?) as they change by season.

As suggested, we have carried out further detailed analyses of the data from BH.

Reviewer #3 (Recommendations for the authors):Figure 1A, the circular histogram of annual catches looks very strange, I think it should be better to show in simple histogram with cartesian coordinates.

We are grateful to reviewer #3 for this valuable suggestion. As indicated above, we have now converted the rose diagram to a simple histogram.

Figure 2B could be removed into the Supplementary, and the sites in Figure 2A also don't need be labeled.

As this work aimed to analyze the migrants collected at BH, we believe that it is important that keep Figure 2A in the main body of the manuscript. Doing so should improve the overall flow of the paper and further accentuate the significance of our work.

Figure 3 is very complicated, I couldn't catch the main idea of this figure. I am not expert on molecular or genetics studies. For me, I am very interesting on the geneflow, and I think Figure 3B and 3D are the most important. I really hope the paper can give more explanation, why the pattern from mitochondrial and nuclear was different? Why the general migration trend towards the Yangtze basin?

As indicated above, we have moved many data (original Panels 3A and 3C) into the Supplementary Information section. We are hopeful that the (multi-panel) figure has thus been simplified. We further emphasize that other studies have also reported (similarly) different patterns for mitochondrial vs. nuclear analyses; we clarify this for future readers in a revised Discussion section (see lines 362-397).

Figure 4 and 5 could be merged together. Firstly, I don't think it is necessary to show all pollen pictures of 46 species. Secondly, map in Figure 5A and 2A is very similar, is it necessary showed it again and again? Third, Figure 5B did not showed very important information and might be better to move it to the Supplementary.

We are hesitant to merge Figure 4 and 5, as they represent markedly different findings (as resulting from different analyses). To help future readers with the interpretation of both figures, we have (slightly) amended Figure legends. Meanwhile, as per reviewer #3’s suggestion, we have moved Figure 5A -B to the Supplementary Information.

Figure 6-8, only a very short paragraph described these three large figures. I think each figure should be simpler, and need more description and more explanation.

As suggested, we have now made substantial revisions to both figures. We also opted to incorporate more information on both figures in the Results (page 8-11, lines 195-259) and Discussion sections (page 16-18, lines 396-452).

[Editors' note: further revisions were suggested prior to acceptance, as described below.]

The manuscript has been improved but there are some remaining issues that need to be addressed, as outlined below.Essential revisions:Please address these three points in your revision and response letter. The individual reviews are provided for your information and will give details to guide your response to the essential revisions, but you do not need to provide a point-by-point response to the reviews – only to the essential revisions.1) Please elaborate on figures 8 and 9 in the results. Currently, even experienced insect migration researchers are still having some difficulty with these figures, so any extra clarification you can provide regarding the take-home message of these figures and their relevance to the story would be helpful. Alternatively, these figures could be moved to supplementary material and briefly summarized in the text.

We agree with the reviewers’ perspective. To avoid confusion among future readers, we have now transferred Figures 8 and 9 to the Supplementary Information.

2) Describe the geographical and biological background of Beihuang Island in the introduction, as the Materials and methods have been moved to the end of the paper.

We are grateful for this valuable comment. We now add this information in the revised draft of our revised manuscript (lines 90-91).